# Privacy-Preserving Federated Convex Optimization: Balancing Partial-Participation and Efficiency via Noise Cancellation

Roie Reshef [*] [1]    Kfir Yehuda Levy [*] [1]

## Abstract

This paper tackles the challenge of achieving Differential Privacy (DP) in Federated Learning (FL) under partial-participation, where only a subset of the machines participate in each time-step. While previous work achieved optimal performance in full-participation settings, these methods struggled to extend to partial-participation scenarios. Our approach fills this gap by introducing a novel noise-cancellation mechanism that preserves privacy without sacrificing convergence rates or computational efficiency. We analyze our method within the Stochastic Convex Optimization (SCO) framework and show that it delivers optimal performance for both homogeneous and heterogeneous data distributions. This work expands the applicability of DP in FL, offering an efficient and practical solution for privacy-preserving learning in distributed systems with partial participation.

## 1. Introduction

Federated Learning (FL) is an innovative framework within Machine Learning (ML) that facilitates collaborative learning across a wide range of decentralized devices or systems (McMahan et al., 2017; Kairouz et al., 2021b). Privacy is a central concern in FL, highlighting the critical importance of safeguarding personal data during the training process.

Differential Privacy (DP) is a strong framework for assessing and mitigating privacy risks (Dwork et al., 2006a;b; Kasiviswanathan et al., 2011). It formally ensures that data analysis results remain largely unchanged regardless of an individual's data inclusion, thus preserving data privacy (Dwork & Roth, 2014).

In this work, we focus on guaranteeing DP in *centralized* FL systems, which are prevalent both in academic research and in many real-world applications. In such systems all machines communicate through a coordinating server which orchestrates the training process (Kairouz et al., 2021b). A key challenge here is that multi-round communication can compromise privacy, and research on integrating DP in FL (Huang et al., 2020; Wei et al., 2020; Girgis et al., 2021; Noble et al., 2022; Lowy & Razaviyayn, 2023; Lowy et al., 2023; Gao et al., 2024) focuses on balancing data masking with model performance to protect privacy while maintaining performance.

While many FL protocols assume *full-participation* from all machines, real-world deployments often face inconsistent device availability, limited connectivity, or scheduling conflicts. This leads to *partial-participation*, where only a subset of machines engage in each training round. Ensuring DP in these settings is more challenging but crucial for expanding FL to large-scale, resource-constrained, and dynamic environments.

Moreover, in the context of privacy, FL systems can be categorized into trusted and untrusted Server cases. In *Trusted Server* scenarios, devices rely on the server's assurance of correct DP implementation. However, in more practical cases with an *Untrusted Server*, devices must protect their data from potential misuse by the server.

Ensuring DP in centralized FL has mainly been studied in the context of Empirical Risk Minimization (ERM) (Huang et al., 2020; Wei et al., 2020; Girgis et al., 2021; Noble et al., 2022), focusing on minimizing training loss. However, translating ERM guarantees to population loss leads to suboptimal bounds, as shown in (Bassily et al., 2019).

Recent works have explored population loss (generalization) guarantees under DP (Lowy & Razaviyayn, 2023; Gao et al., 2024; Reshef & Levy, 2024), providing optimal guarantees for both trusted and untrusted server cases. Notably, (Lowy & Razaviyayn, 2023) addresses both full and partial-participation but relies on mega-batches, leading to super-linear complexity of $O\left(n^{3/2}\right)$, where $n$ is the total dataset size, and (Gao et al., 2024) build upon the algorithm of (Lowy & Razaviyayn, 2023) in iterations to improve the

*Equal contribution [1]Faculty of Electrical and Computer Engineering, Technion, Haifa, Israel. Correspondence to: Roie Reshef <sror@campus.technion.ac.il>, Kfir Yehuda Levy <kfirylevy@technion.ac.il>.

*Proceedings of the 42nd International Conference on Machine Learning*, Vancouver, Canada. PMLR 267, 2025. Copyright 2025 by the author(s).

complexity to $O\left(n^{9/8}\right)$. In contrast, (Reshef & Levy, 2024) achieves linear complexity of $O\left(n\right)$, similar to standard non-private training, but is limited to full-participation.

Despite these advances, a gap remains: *Can we develop a DP-FL method for partial participation that achieves optimal population loss while matching the computational cost of standard training*? In particular, we seek a procedure with overall complexity linear in $n$, the total number of data samples used throughout the training. In this work, we address this gap by proposing a novel approach that ensures near-optimal population-loss guarantees under DP, while preserving the computational efficiency of standard partial-participation methods.

Concretely, in our approach, each participating machine employs exactly one new sample for every round it participates, and use it to performs two stochastic gradient computations, thus keeping the total gradient computations linear in $n$. Thus, if we have a total of $M$ machines but only a subset of $m$ machines ($m \leq M$) participate in each round, for a total of $T$ rounds then overall we employ $n = mT$ samples.

For the trusted-server scenario, we present a simple method that achieves an optimal convergence rate of $O\left(\frac{1}{\sqrt{n}} + \frac{\sqrt{d}}{\epsilon n}\right)$, where $d$ is the problem's dimensionality, and $\epsilon$ is the privacy level. This rate matches the lower bound substantiated in (Bassily et al., 2014). In the more challenging untrusted-server setting, we develop a novel mechanism that yields an optimal convergence rate of $O\left(\frac{1}{\sqrt{n}} + \frac{\sqrt{Md}}{\epsilon n}\right)$ where $M$ is the total number of machines. This matches the lower bound substantiated in (Lowy & Razaviyayn, 2023).

Our work builds on the full-participation approach of (Reshef & Levy, 2024). To address the challenging untrusted server and partial-participation case, we introduce a novel noise-cancellation mechanism that effectively obscures the information sent to the server, rather than the independent noise injection employed by (Reshef & Levy, 2024). Additionally, we tailor each machine's noise injection based on the number of rounds it participates in, thus balancing privacy and performance.

**Related Work.** Several studies have made significant contributions to DP in the context of SCO, particularly in ERM (Chaudhuri et al., 2011; Kifer et al., 2012; Thakurta & Smith, 2013; Song et al., 2013; Duchi et al., 2013; Ullman, 2015; Talwar et al., 2015; Wu et al., 2017; Wang et al., 2018; Iyengar et al., 2019; Kairouz et al., 2021a; Avella-Medina et al., 2021; Ganesh et al., 2023). However, these works mainly focus on training loss, and attempting to extend their results to population loss guarantees through standard uniform convergence (Shalev-Shwartz et al., 2009) results in sub-optimal bounds, as discussed in (Bassily et al., 2019; Feldman et al., 2020).

Structured noise for DP has been explored in prior work. The work of (Koloskova et al., 2023) suggests adding correlated noise in ERM settings using standard gradients, but without utility guarantees or support for partial-participation. Moreover, the work of (Hafeez et al., 2021), while also using noise cancellation, addresses smart metering rather than ML and offers no formal guarantees.

The work of (Bassily et al., 2014) established a lower bound for DP-SCO, and (Bassily et al., 2019) presented an algorithm that achieves this bound, though with super-linear computational complexity $\propto n^{3/2}$. This was later improved by (Feldman et al., 2020), which achieved optimal guarantees with a sample complexity linear in $n$, but only provides privacy for the final iterate, making it unsuitable to be extended to FL settings.

In the FL setting, a notable work is (Cheu et al., 2022), which provides the optimal bounds for the trusted server case, though it relies on an expensive vector-shuffling routine, resulting in a computational complexity greater than $n^{3/2}$. Another important work is (Lowy & Razaviyayn, 2023), which combines the large-batch technique of (Bassily et al., 2019) with other mechanisms to achieve optimal results for both untrusted server (called ISRL-DP in their work) and trusted server (called SDP in their work) cases, operating in the partial-participation setting with a computational complexity of $n^{3/2}$, and (Gao et al., 2024) further improving it to $n^{9/8}$. However, the works of (Lowy & Razaviyayn, 2023; Gao et al., 2024) achieve the optimal bound only in the full-participation setting, while suffering an extra multiplicative factor of $\sqrt{\frac{M}{m}}$ to their excess loss bound in the partial-participation setting.

Our approach draws on the private extension of (Levy, 2023) introduced in (Reshef & Levy, 2024), which achieves both optimal convergence and computational efficiency under trusted and untrusted server cases, but only in a full-participation setting. We extend these results to the more practical partial-participation setting, retaining the same tight population-loss guarantees and linear computational complexity.

## 2. Preliminaries

Here we provide the necessary background for analyzing private federated learning.
**Notations:** We will employ the following notations: $[n] := \{1, \ldots, n\}$, $\alpha_{1:t} := \sum_{\tau=1}^{t} \alpha_\tau$, and $\Pi_\mathcal{K}\left(x\right) := \arg\min_{y \in \mathcal{K}} \|x - y\|$.

### 2.1. Federated Learning

We focus on Stochastic Convex Optimization (SCO) scenarios where we have $M$ different machines, and each machine

$i \in [M]$ has a convex objective function, $f_i : \mathcal{K} \mapsto \mathbb{R}$, which takes the following form:

$$f_i(x) = \mathbb{E}_{z \sim \mathcal{D}_i} [f_i(x; z)] \tag{1}$$

Where $\mathcal{K} \subset \mathbb{R}^d$ is a compact convex set, and $\mathcal{D}_i$ is an unknown data distribution, from which machine $i$ may draw i.i.d. samples. Our goal is minimizing the average objective:

$$f(x) := \frac{1}{M} \sum_{i=1}^{M} f_i(x) \tag{2}$$

In federated learning the machines aim to collaboratively minimize the above objective, where each machine $i$ may acess a dataset of $\mathcal{S}_i = \{z_1, \ldots, z_{n_i}\} \subset \mathcal{Z}_i$ ($\mathcal{Z}_i$ is the set where the samples of machine $i$ reside), of $n_i$ i.i.d. samples from $\mathcal{D}_i$. Upon utilizing these data samples the learning algorithm eventually outputs a solution $x_{\text{out}} \in \mathcal{K}$. Our performance metric is the expected excess loss $\mathcal{R}(x_{\text{out}})$:

$$\mathcal{R}(x_{\text{out}}) := \mathbb{E} [f(x_{\text{out}})] - \min_{x \in \mathcal{K}} \{f(x)\} \tag{3}$$

The expectation is w.r.t. the randomness of the samples, as well as w.r.t. the (possible) randomization of the algorithm.

We focus on centralized systems where machines can synchronize and communicate through a central entity called the *Parameter Server* ($\mathcal{PS}$), which orchestrates the learning process. We further focus on *Partial-Participation* scenarios where at each communication round $t$ only a random subset of the machines $\mathcal{M}_t \subseteq [M]$ communicate with the $\mathcal{PS}$.

We focus on the common parallelization scheme inspired by Minibatch-SGD (Dekel et al., 2012). In this scheme, the $\mathcal{PS}$ maintains a sequence of query points $\{x_t\}_t$ which are updated using gradient information that is gathered from the participating machines. Concretely, at time step $t$ the $\mathcal{PS}$ sends the query point $x_t$ to a set of randomly chosen machines $\mathcal{M}_t \subseteq [M]$. Each machine $i \in \mathcal{M}_t$ draws a new sample $z_{t,i} \sim \mathcal{D}_i$ (taken from $\mathcal{S}_i$), independently of past samples, and calculates a gradient estimate $g_{t,i} = \nabla f_i(x_t; z_{t,i})$, where the derivative is with respect to $x$. The $\mathcal{PS}$ aggregates the gradient estimates into $g_t = \frac{1}{|\mathcal{M}_t|} \sum_{i \in \mathcal{M}_t} g_{t,i}$, which is then used to compute the next query point $x_{t+1}$

The independence of the samples guarantees that $g_{t,i}$ is an unbiased estimator of $\nabla f_i(x_t)$, meaning that $\mathbb{E}[g_{t,i}|x_t] = \nabla f_i(x_t)$. It is useful to conceptualize the calculation of $g_{t,i} = \nabla f_i(x_t; z_{t,i})$ as a sort of (noisy) Gradient Oracle: upon receiving a query point $x_t \in \mathcal{K}$, this Oracle outputs a vector $g_{t,i} \in \mathbb{R}^d$, serving as an unbiased estimate of $\nabla f_i(x_t)$. Moreover, assuming $\mathcal{M}_t$ is picked uniformly over the $M$ machines, directly implies that $g_t = \frac{1}{|\mathcal{M}_t|} \sum_{i \in \mathcal{M}_t} g_{t,i}$, is an unbiased estimate of $\nabla f(x_t)$.

**Assumptions:** We will make the following assumptions:
**Diameter:** $\exists D > 0$ such: $\|x - y\| \leq D, \ \forall x, y \in \mathcal{K}$.

We also make assumptions about $f_i(\cdot; z), \forall i \in [M], z \in \mathcal{Z}_i$:
**Lipschitz:** $\exists G > 0$ such:

$$|f_i(x; z) - f_i(y; z)| \leq G \|x - y\|, \quad \forall x, y \in \mathcal{K}$$

This also implies that $\|\nabla f_i(x; z)\| \leq G, \forall x \in \mathcal{K}$.
**Smoothness:** $\exists L > 0$ such:

$$\|\nabla f_i(x; z) - \nabla f_i(y; z)\| \leq L \|x - y\|, \quad \forall x, y \in \mathcal{K}$$

Since these assumptions hold for $f_i(x; z)$ for every $i \in [M], z \in \mathcal{Z}_i$, they also hold for $f_i(x)$ and $f(x)$. These assumptions imply the following (proof in Appendix B.1):
**Bounded Variance:** $\exists \sigma \in [0, G]$ such:

$$\mathbb{E} \|\nabla f_i(x; z) - \nabla f_i(x)\|^2 \leq \sigma^2, \quad \forall x \in \mathcal{K} \tag{4}$$

**Bounded Smoothness Variance:** $\exists \sigma_L \in [0, L]$ such:

$$\mathbb{E} \|(\nabla f_i(x; z) - \nabla f_i(x)) - (\nabla f_i(y; z) - \nabla f_i(y))\|^2$$
$$\leq \sigma_L^2 \|x - y\|^2, \quad \forall x, y \in \mathcal{K} \tag{5}$$

**Bounded Heterogeneity:** $\exists \xi \in [0, G]$ such:

$$\frac{1}{M} \sum_{i=1}^{M} \|\nabla f_i(x) - \nabla f(x)\|^2 \leq \xi^2, \forall x \in \mathcal{K} \tag{6}$$

**Bounded Smoothness Heterogeneity:** $\exists \xi_L \in [0, L]$ such:

$$\frac{1}{M} \sum_{i=1}^{M} \|(\nabla f_i(x) - \nabla f(x)) - (\nabla f_i(y) - \nabla f(y))\|^2$$
$$\leq \xi_L^2 \|x - y\|^2, \quad \forall x, y \in \mathcal{K} \tag{7}$$

These properties allow us to bound the variance of the average of the gradients (proof in Appendix B.2):

**Lemma 2.1.** *Let $\mathcal{M} \subseteq [M]$, and $z_i \sim \mathcal{D}_i$. Define $g(x) = \frac{1}{|\mathcal{M}|} \sum_{i \in \mathcal{M}} \nabla f_i(x; z_i)$, for all $x \in \mathcal{K}$. If $z_i$ are independent, and $\mathcal{M}$ is chosen uniformly from all subsets of size $m$:*

$$\mathbb{E} \|g(x) - \nabla f(x)\|^2 \leq \frac{1}{m} \left( \sigma^2 + \frac{M - m}{M - 1} \xi^2 \right), \quad \forall x \in \mathcal{K}$$

$$\mathbb{E} \|(g(x) - \nabla f(x)) - (g(y) - \nabla f(y))\|^2$$
$$\leq \frac{1}{m} \left( \sigma_L^2 + \frac{M - m}{M - 1} \xi_L^2 \right) \|x - y\|^2, \quad \forall x, y \in \mathcal{K}$$

Under these assumptions, $g(x)$ is an unbiased estimate of $\nabla f(x)$. When $m = M$, heterogeneity disappears, and as $m$ gets smaller relative to $M$, heterogeneity becomes more significant. Additionally, since $\frac{M-m}{M-1} \leq 1$, we can use the bounds $\sigma^2 + \xi^2$ and $\sigma_L^2 + \xi_L^2$. Thus, heterogeneity in partial participation adds an extra term to the variance. In this work, we assume the number of participating machines per round is fixed, and denote it by $m$.

## 2.2. Differential Privacy

The core concept of Differential Privacy (DP) is to compare the algorithm's output with and without private information and quantify the difference between these outputs. The closer they are, the more private the algorithm is. This is measured by the difference between the probability distributions of the outputs.

The **Rényi Divergence** is a popular difference measure between probability distributions:

**Definition 2.2** (Rényi Divergence (Rényi, 1961)). Let $P, Q$ be probability distributions over the same set, and let $\alpha > 1$. The Rényi divergence of order $\alpha$ between $P$ and $Q$ is:

$$\mathbb{D}_\alpha\left(P\|Q\right) := \frac{1}{\alpha - 1}\log\left(\mathbb{E}_{X \sim P}\left[\left(\frac{P(X)}{Q(X)}\right)^{\alpha - 1}\right]\right)$$

We follow with the convention that $\frac{0}{0} = 0$. If $Q(x) = 0$, but $P(x) \neq 0$ for some $x$, then the Rényi divergence is defined to be $\infty$. Divergence of orders $\alpha = 1, \infty$ are defined by continuity.

**Notation:** If $X \sim P, Y \sim Q$, we will use the terms $\mathbb{D}_\alpha\left(P\|Q\right)$ & $\mathbb{D}_\alpha\left(X\|Y\right)$ interchangeably.

Adding Gaussian noise is popular in privacy, so we find the following lemma useful (proof in Appendix B.3):

**Lemma 2.3.** *Let $P = \mathcal{N}(\mu, I\sigma^2)$ and $Q = \mathcal{N}(\mu + \Delta, I\sigma^2)$, two Gaussian distributions. Then, $\mathbb{D}_\alpha\left(P\|Q\right) = \frac{\alpha\|\Delta\|^2}{2\sigma^2}$.*

The classic differential privacy definition:

**Definition 2.4** (Differential Privacy (Dwork et al., 2006a;b)). A randomized algorithm $\mathcal{A}$ is $(\epsilon, \delta)$-differentially private, or $(\epsilon, \delta)$-DP, if for all neighboring datasets $\mathcal{S}, \mathcal{S}'$ that differs in a single element, and for all events $\mathcal{O}$:

$$\mathbb{P}\left\{\mathcal{A}(\mathcal{S}) = \mathcal{O}\right\} \leq e^\epsilon \mathbb{P}\left\{\mathcal{A}(\mathcal{S}') = \mathcal{O}\right\} + \delta$$

The term *neighboring datasets* refers to $\mathcal{S}, \mathcal{S}'$ being *ordered* sets of data samples that only differ on *a single sample*.

A privacy measure that relies on the Rényi divergence:

**Definition 2.5** (Rényi Differential Privacy (Mironov, 2017)). For $1 \leq \alpha \leq \infty$ and $\epsilon \geq 0$, a randomized algorithm $\mathcal{A}$ is $(\alpha, \epsilon)$-Rényi differentially private, or $(\alpha, \epsilon)$-RDP, if for all neighboring datasets $\mathcal{S}, \mathcal{S}'$: $\mathbb{D}_\alpha\left(\mathcal{A}(\mathcal{S})\|\mathcal{A}(\mathcal{S}')\right) \leq \epsilon$.

RDP implies DP, as shown here (proof in Appendix B.4):

**Lemma 2.6** ((Mironov, 2017)). *If $\mathcal{A}$ satisfies $(\alpha, \epsilon)$-RDP, then for all $\delta \in (0, 1)$, it also satisfies $\left(\epsilon + \frac{\log\left(\frac{1}{\delta}\right)}{\alpha - 1}, \delta\right)$- DP. In particular, if $\mathcal{A}$ satisfies $\left(\alpha, \frac{\alpha\rho^2}{2}\right)$-RDP for every $\alpha > 1$, then for all $\delta \in (0, 1)$, it also satisfies $\left(\frac{\rho^2}{2} + \rho\sqrt{2\log\left(\frac{1}{\delta}\right)}, \delta\right)$-DP.*

Lastly, for the case of Gaussian noise, there is a more fitting DP definition based on RDP:

**Definition 2.7** (Zero-Concentrated Differential Privacy). (Bun & Steinke, 2016) A randomized algorithm $\mathcal{A}$ is $(\xi, \rho)$-zero-concentrated differentially private, or $(\xi, \rho)$-zCDP, if for all neighboring datasets $\mathcal{S}, \mathcal{S}'$ that differ in a single element, and for all $\alpha > 1$, we have:

$$\mathbb{D}_\alpha\left(\mathcal{A}(\mathcal{S})\|\mathcal{A}(\mathcal{S}')\right) \leq \xi + \rho\alpha$$

We define $\rho$-zCDP as $(0, \rho)$-zCDP.

Note that if an algorithm is $(\alpha, \xi + \rho\alpha)$-RDP for all $\alpha > 1$, it is also $(\xi, \rho)$-zCDP and visa-versa. Also note that according to Lemma 2.3, a Gaussian mechanism is $\frac{\|\Delta\|^2}{2\sigma^2}$-zCDP, and using Lemma 2.6, if an algorithm is $\frac{\rho^2}{2}$-zCDP, it is also $\left(\frac{\rho^2}{2} + \rho\sqrt{2\log\left(\frac{1}{\delta}\right)}, \delta\right)$-DP, for all $\delta \in (0, 1)$. By choosing to work with $\frac{\rho^2}{2}$-zCDP, we get that $\rho = \frac{\|\Delta\|}{\sigma}$, which is the signal-to-noise ratio, and we also get that $O\left(\frac{1}{\rho}\right) \in O\left(\frac{1}{\epsilon}\right)$, with $\epsilon = \frac{\rho^2}{2} + \rho\sqrt{2\log\left(\frac{1}{\delta}\right)}$.

In federated learning, we consider the case where DP guarantees must be ensured individually for each machine $i \in [M]$. The challenge in this context arises because each machine communicates multiple times with the $\mathcal{PS}$, potentially revealing its private dataset. The machines don't trust one another, and thus cannot allow them to uncover private data. In this work, we mainly focus on the **Untrusted Server** case, where the $\mathcal{PS}$ cannot reveal private data, and DP guarantees must be ensured for the information machines share with the $\mathcal{PS}$. In the **Trusted Server** case, machines trust the $\mathcal{PS}$ and may share private data, but it is still necessary to prevent machines from uncovering each other's private data from the $\mathcal{PS}$.

## 2.3. Lower Bounds

Here we elaborate on the existing excess loss lower bounds for the DP trusted and untrusted server scenarios, and show that they coincide with the upper bounds that we establish. Notably, existing lower bounds holds irrespective of the training method, and therefore apply simultaneously for both full and partial-participation settings. Concretely, the starting points of such lower bounds is to assume that the learning process may access a total of $n$ data points.

In the **Trusted Server** setting we can think of the server as learner that may access $n$ data points, and aims to output a solution that maintains a DP level of $\epsilon$. The work of (Bassily et al., 2014) establishes a lower bound of $\Omega\left(\frac{1}{\sqrt{n}} + \frac{\sqrt{d}}{\epsilon n}\right)$ on the excess loss in this case, which matches the term in our upper bound which is related to privacy.

In the **Untrusted Server** setting we have $M$ machines,

which aim to collaboratively compute a solution to a given stochastic optimization problem. Yet each machine and aims to maintains a DP level of $\epsilon$ for the data that it shares during the learning process. It is also assume that the overall data used by all machines is of size $n$. The work of (Lowy & Razaviyayn, 2023) establishes a lower bound of $\Omega\left(\frac{1}{\sqrt{n}} + \frac{\sqrt{Md}}{\epsilon n}\right)$ on the excess loss in this case, which matches the term in our upper bound which is related to privacy. The extra $\sqrt{M}$ factor exists because we have $M$ devices that hold privacy, instead of just one (the $\mathcal{PS}$).

Note that the first term in our lower bounds $\Omega\left(\frac{1}{\sqrt{n}}\right)$ matches the known lower and upper bounds of excess loss in non-DP setting. Finally, note that in our partial-participation setting, only a subset of $m < M$ machines participate once in each round, for a total of $T$ rounds, so we have $n = mT$.

## 3. The $\mu^2$ Technique

As mentioned earlier, our technique for the partial-participation case builds on the previous approach of (Reshef & Levy, 2024) for the full-participation case. Here we discuss the algorithmic techniques behind the latter, which will later serve us in our design of an optimal and efficient approach for partial-participation.

The approach of (Reshef & Levy, 2024) strongly relies on a recent (non-DP) algorithmic approach called $\mu^2$-SGD (Levy, 2023). Next we elaborate on it.

### 3.1. The $\mu^2$-SGD Algorithm

The $\mu^2$-SGD (Levy, 2023) is a variant of standard SGD with two modifications. Its update rule is of the following form: $w_1 = x_1 \in \mathcal{K}$, and $\forall t \geq 1$:

$$w_{t+1} = \Pi_{\mathcal{K}}\left(w_t - \eta \alpha_t d_t\right)$$
$$x_{t+1} = \frac{\alpha_{1:t}}{\alpha_{1:t+1}} x_t + \frac{\alpha_{t+1}}{\alpha_{1:t+1}} w_{t+1} \quad (8)$$

Here $\alpha_t > 0$ are importance weights for each time-step. We use $\alpha_t \propto t$, giving more weight to more recent updates.

Note that this update rule maintains two sequences: $\{w_t\}_t$, $\{x_t\}_t$, where $\{x_t\}_t$ is a sequence of weighted averages of the iterates $\{w_t\}_t$. $d_t$ is an estimate for the gradient at the *average point*, i.e. of $\nabla f(x_t)$, which differs from standard SGD which employs estimates for the gradients *at the iterates*, i.e. of $\nabla f(w_t)$. This approach is related to a technique called Anytime-GD (Cutkosky, 2019), which is strongly connected to the notions of momentum and acceleration (Cutkosky, 2019; Kavis et al., 2019).

In the standard SGD version of Anytime-GD, one would use the estimate $\nabla f(x_t; z_t)$. However, the $\mu^2$-SGD approach suggests to employ a variance reduction mechanism

to yield a *corrected momentum* estimate $d_t$ in the spirit of (Cutkosky & Orabona, 2019), with a technique called Stochastic Corrected Momentum (STORM). This is done as follows: $d_1 := \nabla f(x_1; z_1)$, and $\forall t > 1$:

$$d_t = \nabla f(x_t; z_t) + (1 - \beta_t)(d_{t-1} - \nabla f(x_{t-1}; z_t)) \quad (9)$$

Where $\beta_t \in [0, 1]$ are called *corrected momentum* weights. It can be shown by induction that $\mathbb{E}[d_t] = \mathbb{E}[\nabla f(x_t)]$, but generally $\mathbb{E}[d_t|x_t] \neq \nabla f(x_t)$, in contrast to standard SGD estimators. However, as demonstrated in (Levy, 2023) by choosing corrected momentum weights of $\beta_t \propto \frac{1}{t}$, the above estimates achieve an error reduction at time-step $t$ of:

$$\mathbb{E}\left\|\varepsilon_t^{\mu^2}\right\|^2 := \mathbb{E}\|d_t - \nabla f(x_t)\|^2 \leq O\left(\frac{1}{mt}\left(\tilde{\sigma}^2 + \tilde{\xi}^2\right)\right)$$

Where $\tilde{\sigma}^2 \leq O\left(\sigma^2 + \sigma_L^2 D^2\right)$ and $\tilde{\xi}^2 \leq O\left(\xi^2 + \xi_L^2 D^2\right)$. This implies that the error decreases with $t$, in contrast to standard SGD where the variance $\mathbb{E}\left\|\varepsilon_t^{\text{SGD}}\right\|^2 := \mathbb{E}\|g_t - \nabla f(x_t)\|^2$ remains uniformly bounded by $\frac{1}{m}\left(\sigma^2 + \xi^2\right)$, as shown in Lemma 2.1.

Upon choosing $\beta_t = 1 - \frac{\alpha_{t-1}}{\alpha_t}$, and denoting $q_t := \alpha_t d_t$, the gradient estimate update of $\mu^2$-SGD in Equation (9) can be written as follows (proof in Appendix C.1):

$$s_t := \alpha_t \nabla f(x_t; z_t) - \alpha_{t-1}\nabla f(x_{t-1}; z_t)$$
$$q_t = q_{t-1} + s_t \quad (10)$$

### 3.2. Differentially Private $\mu^2$ Federated Learning

The $\mu^2$-SGD approach is naturally extends to the FL setting as follows: At every time-step, $t$ the $\mathcal{PS}$ sends the current query point $x_t$ to *all machines*. Then, each machine $i \in [M]$ updates a local estimate of the (weighted) gradient $q_{t,i} = \alpha_t d_{t,i}$ based on its local (and private) data and on $x_t$. This is done similarly to Equation (10), with $z_{t,i}$ being the sample used by machine $i$ at time-step $t$:

$$s_{t,i} := \alpha_t \nabla f(x_t; z_{t,i}) - \alpha_{t-1}\nabla f(x_{t-1}; z_{t,i})$$
$$q_{t,i} = q_{t-1,i} + s_{t,i} \quad (11)$$

In the **trusted server** case, (Reshef & Levy, 2024) suggests to aggregate these estimates by the $\mathcal{PS}$ into $q_t := \frac{1}{M}\sum_{i=1}^{M} q_{t,i}$. To maintain privacy, the server adds a fresh noise $Y_t$ into $q_t$ and use $\tilde{q}_t := q_t + Y_t$ (a common technique in DP training (Abadi et al., 2016)). In the **untrusted server** case, (Reshef & Levy, 2024) suggests adding a fresh zero-mean noise $Y_{t,i}$ to the gradient estimates $q_{t,i}$, in order to ensure privacy, and then send $\tilde{q}_{t,i} := q_{t,i} + Y_{t,i}$ to the $\mathcal{PS}$. It aggregate these private estimates into $\tilde{q}_t = \frac{1}{M}\sum_{i=1}^{M} \tilde{q}_{t,i}$.

In both case, the $\mathcal{PS}$ then updates $w_t$ and $x_t$ similarly to Equation (8), while using $\tilde{q}_t$ instead of $\alpha_t d_t$. Upon choosing appropriate noise magnitude and learning rates, this

approach ensures optimal convergence guarantees for DP training in full-participation settings, while preserving the linear computational complexity of standard methods.

There is a natural trade-off in selecting the noise magnitude: larger noise improves privacy but slows convergence. As shown in (Reshef & Levy, 2024), the $\mu^2$-SGD algorithm is substantially less sensitive to individual data samples compared to standard SGD, so we can maintain privacy with relatively small noise magnitudes, thus achieving optimal convergence.

## 4. Our Approach

Here we discuss the challenges of extending DP-$\mu^2$-FL to Partial-Participation scenarios (Section 4.1), and suggest a natural modification that remedies these challenges in the *Trusted server* case, thus leading to optimal and computationally efficient methods for this case. Unfortunately, this modification leads to suboptimal performance in the more challenging *Untrusted server* case.

Towards addressing this gap, in Section 4.2 we provide a new perspective on DP-$\mu^2$-FL in the full-participation case, which induces a novel noise-cancellation technique, allowing us to extend their method to Partial-Participation settings. Finally, in Section 4.3 we discuss the properties of the resulting gradient estimates, which are later used in to analyze our approach. In Section 5 we present our complete algorithm (Algorithm 1) and its guarantees.

### 4.1. Challenge in Partial-Participation

The performance of DP-$\mu^2$-FL strongly relies on the assumption that *every* machine can access the *entire* sequence of query points $\{x_t\}_t$, which is crucial for computing $q_{t,i}$ (see Equation (11)). Unfortunately, this assumption no longer holds in partial-participation setting.

**Alternative 1:** A natural resolution is to update $q_{t,i}$ based on the sequence of queries that machine $i$ may access:

$$q_{t,i} = q_{t-\tau,i} + \alpha_t \nabla f(x_t; z_{t,i}) - \alpha_{t-\tau} \nabla f(x_{t-\tau}; z_{t,i}) \tag{12}$$

Where $t - \tau$ is the previous time-step that machine $i$ participated in. Unfortunately, this approach yields suboptimal guarantees even in standard non-DP partial-participation settings. The delayed updates lead to an excessive error for $q_{t,i}$, resulting a degraded convergence rate of $O\left(\sqrt{\frac{M}{m}}\frac{1}{\sqrt{n}}\right)$ even without DP requirements. We elaborate on it in Appendix G.

**Alternative 2:** Another natural solution is to simply let the $\mathcal{PS}$ calculate $q_t$. This suggest the following approach for the partial-participation setting: A machine $i$ participating at round $t$ computes $s_{t,i}$ (Equation (11)), and sends it the the $\mathcal{PS}$, who averages them into $s_t = \frac{1}{m}\sum_{i\in\mathcal{M}_t} s_{t,i}$ and

uses these estimates to update $q_t$ (Equation (10)).

This approach works well in the *Trusted server* case, where the $\mathcal{PS}$ can receive the non-private $s_{t,i}$ from the participating machines, and use them to update $q_t$. Similarly to the approach of (Reshef & Levy, 2024) the $\mathcal{PS}$ then injects noise into $q_t$ to maintain its privacy, and then updates according to Equation (8). We describe this approach in Appendix F, and substantiate a convergence rate of $O\left(\frac{1}{\sqrt{n}} + \frac{\sqrt{d}}{\epsilon n}\right)$.

Nevertheless, extending this approach to the *Untrusted server* case is not trivial. Concretely, one can think of the natural extension where each machine $i$ that participates at round $t$ computes a private estimate of $s_{t,i}$, and then privatize it by adding a fresh noise $\tilde{s}_{t,i} := s_{t,i} + Y_{t,i}$ which is then sent to the $\mathcal{PS}$. The latter utilizes the private $\tilde{s}_{t,i}$ to update $\tilde{q}_t$ which is then used to update in the spirit of Equation (8). Unfortunately, ensuring the privacy of $\tilde{s}_{t,i}$ that way, leads to a cumulative noise in the global estimate $\tilde{q}_t$, which leads to highly suboptimal performance. Next, we will see how to mend this approach by injecting *correlated* noise to $s_{t,i}$, which leads to noise cancellation at $\tilde{q}_t$, thus enabling to achieve optimal performance.

### 4.2. Noise Cancellation

We focus on addressing the untrusted server partial-participation case. Towards doing so, we provide an alternative perspective on the approach of (Reshef & Levy, 2024) for the full-participation. In the full-participation case, Equation (11) implies $q_{t,i} = \sum_{\tau=1}^{t} s_{\tau,i}$. Since $\tilde{q}_{t,i} = q_{t,i} + Y_{t,i}$, we can write the update rule for $\tilde{q}_t$ in the $\mathcal{PS}$ like this:

$$\tilde{q}_t = \tilde{q}_{t-1} + s_t + Y_t - Y_{t-1} := \tilde{q}_{t-1} + \tilde{s}_t$$

Where $s_t := \frac{1}{M}\sum_{i=1}^{M} s_{t,i}$, $Y_t := \frac{1}{M}\sum_{i=1}^{M} Y_{t,i}$, and $\tilde{s}_t := s_t + Y_t - Y_{t-1}$. The above update rule for $\tilde{q}_t$ can be equivalently achieved as follows: at every round $t$ each machine $i$ computes $\tilde{s}_{t,i} = s_{t,i} + Y_{t,i} - Y_{t-1,i}$, which is equivalent to injecting $s_t$ with a fresh noise term while canceling the previous one. Then each machine sends $\tilde{s}_{t,i}$ to the $\mathcal{PS}$, which aggregates them into $\tilde{s}_t = \frac{1}{M}\sum_{i=1}^{M} \tilde{s}_{t,i}$, and updates the estimate $\tilde{q}_t = \tilde{q}_{t-1} + \tilde{s}_t$. The sequence of noise terms that we inject into $s_{t,i}$, $\{Y_{t,i} - Y_{t-1,i}\}_t$, is now correlated.

This perspective now induces the following approach for the partial-participation setting: at each time-step $t$, only some of the machines, $\mathcal{M}_t \subseteq [M]$, are participating. We employ this update rule (Algorithm 1):

$$
\begin{aligned}
s_{t,i} &= \alpha_t \nabla f(x_t; z_{t,i}) - \alpha_{t-1} \nabla f(x_{t-1}; z_{t,i}) \\
\tilde{s}_{t,i} &= s_{t,i} + Y_{t,i} - Y_{t-1,i} \\
\tilde{s}_t &= \frac{1}{m}\sum_{i\in\mathcal{M}_t} \tilde{s}_{t,i} \quad \& \quad \tilde{q}_t = \tilde{q}_{t-1} + \tilde{s}_t
\end{aligned}
\tag{13}
$$

And the $\mathcal{PS}$ then uses $\tilde{q}_t$ to update $w_t$ and $x_t$ as in Equation (8). The noise $Y_{t,i}$ being:

$$Y_{t,i} = \begin{cases} y_{t,i} & i \in \mathcal{M}_t \\ Y_{t-1,i} & i \notin \mathcal{M}_t \end{cases} \tag{14}$$

Where the noise $y_{t,i} \sim P_{t,i}$ is the new independent noise generated at time-step $t$ in machine $i$, and $Y_{t,i}$ is the last noise generated up to time-step $t$ in machine $i$. As we can see, for the machines that participate at time-step $t$, we generate a new noise, and the machines that do not participate at time-step $t$ retain their previous noise. Using this noise, we can see how $\tilde{q}_t$ looks like (proof in Appendix D.1):

**Lemma 4.1.** *Our definitions of $\tilde{q}_t, q_t, Y_{t,i}$ above imply:*

$$\tilde{q}_t = q_t + \frac{1}{m}\sum_{i=1}^{M} Y_{t,i}$$

Due to our noise cancellation, the effective injected noise on the $\mathcal{PS}$ at time $t$ is $Y_t = \frac{1}{m}\sum_{i=1}^{M} Y_{t,i}$. Note that $Y_t$ is a sum a total of $M$ different independent noises (one from each machine), but normalized by the number of machines participating at each round $m$. Also note that since for some machines $Y_{t,i} = Y_{t-1,i}$, the noises of different time-steps are not independent. These two points will make it harder for us to evaluate the effect of this noise on the excess loss.

### 4.3. Gradient Error & Sensitivity Analysis

Here we analyze the properties of the $s_{t,i}$ estimates depicted above, as well as discuss the properties of $q_t$. These properties are crucial for analyzing the privacy and performance of our approach. Our analysis follows similar lines to (Levy, 2023; Reshef & Levy, 2024).

We define: $g_{t,i} := \nabla f_i(x_t; z_{t,i})$, $\tilde{g}_{t,i} := \nabla f_i(x_t, z_{t+1,i})$, $\bar{g}_{t,i} := \nabla f_i(x_t)$, $\bar{g}_t := \nabla f(x_t)$. By these notation we can write $s_{t,i} = \alpha_t g_{t,i} - \alpha_{t-1}\tilde{g}_{t-1,i}$. We will also use the notation $\mathbb{E}_{t-1}[\cdot]$ to denote the expectation conditioned over all randomization up to time-step $t-1$. Using this notation we get that: $\mathbb{E}_{t-1}[s_{t,i}] = \alpha_t \bar{g}_{t,i} - \alpha_{t-1}\bar{g}_{t-1,i}$. We define $\bar{s}_{t,i} := \alpha_t \bar{g}_{t,i} - \alpha_{t-1}\bar{g}_{t-1,i}$, and $\bar{s}_t = \alpha_t \bar{g}_t - \alpha_{t-1}\bar{g}_{t-1}$. Next we bound $s_{t,i}$ (proof in Lemma 4.2):

**Lemma 4.2** ((Reshef & Levy, 2024)). *Let $\mathcal{K} \subset \mathbb{R}^d$ be a convex set of diameter $D$, and $\{f_i(\cdot; z)\}_{z \in \mathcal{Z}_i}$ be a family of convex $G$-Lipschitz and $L$-smooth functions. Assume $\alpha_t = t$ and define $S := G + 2LD, \tilde{\sigma} := \sigma + 2\sigma_L D, \tilde{\xi} := \xi + 2\xi_L D$:*

$$\|s_{t,i}\| \le S, \quad and \quad \mathbb{E}\|s_t - \bar{s}_t\|^2 \le \frac{1}{m}\left(\tilde{\sigma}^2 + \frac{M-m}{M-1}\tilde{\xi}^2\right)$$

*Where the in expectation bound further assumes that the samples in $\mathcal{Z}_i$ arrive from i.i.d. $\mathcal{D}_i$.*

Now, we will define the error of of our weighted corrected momentum estimate $\varepsilon_t := q_t - \alpha_t \nabla f(x_t)$. Note that the update rule for $\varepsilon_t$ is (proof in Appendix D.3):

$$\varepsilon_t = \varepsilon_{t-1} + (s_t - \bar{s}_t) \tag{15}$$

And the above implies $\varepsilon_t = \sum_{\tau=1}^{t}(s_\tau - \bar{s}_\tau)$. The above enable to bound the error $\varepsilon_t$ (proof in Appendix D.4):

**Lemma 4.3** ((Reshef & Levy, 2024)). *Algorithm 1 with weights $\alpha_t = t$ ensures:*

$$\mathbb{E}\|\varepsilon_t\|^2 := \mathbb{E}\|q_t - \alpha_t \nabla f(x_t)\|^2 \le \frac{t}{m}\left(\tilde{\sigma}^2 + \frac{M-m}{M-1}\tilde{\xi}^2\right)$$

## 5. DP-$\mu^2$-FL with Partial-Participation

Our complete algorithm depicted in Algorithm 1, is based on the noise cancellation idea we describe in Equations (13) and (14), combined with the $\mu^2$-SGD approach described in Equation (8). At each time-step $t$, we select a subset of $m$ machines $\mathcal{M}_t$. Each machine $i \in \mathcal{M}_t$ computes the gradient estimate correction $s_{t,i}$, and then adds a fresh noise term $y_{t,i}$ and removes the previous noise $Y_i := Y_{t-1,i}$ (omitting the time-step index for practical implementation) to get a private correction $\tilde{s}_{t,i}$. It is transmitted to the $\mathcal{PS}$, which averages these corrections, updates its weighted noisy gradient estimate $\tilde{q}_t$, and in turn uses $\tilde{q}_t$ to compute the next iterate $w_{t+1}$ and query point $x_{t+1}$.

To balance privacy with performance, we found it necessary to set each machine's noise injection proportionally to the number of time-steps it participated in up until now, because a machine that participates more rounds may leak more private data, which requires more noise to maintain its privacy. Thus, we set $\sigma_{t,i}^2 \propto N_{t,i}$, where $N_{t,i}$ is the number of time-steps machine $i$ participated up to time step $t$.

### 5.1. Privacy Guarantees

We shows how the privacy of Algorithm 1 depends on the injected noises $\{y_{t,i}\}$ (proof in Appendix E.1):

**Theorem 5.1.** *Let $\mathcal{K} \subset \mathbb{R}^d$ be a convex set of diameter $D$, and $\{f_i(\cdot; z)\}_{z \in \mathcal{Z}_i}$ be a family of convex $G$-Lipschitz and $L$-smooth functions, and $S = G + 2LD$. Then invoking Algorithm 1 with noise distributions $y_{t,i} \sim P_{t,i} = \mathcal{N}\left(0, I\sigma_{t,i}^2\right)$, and any learning rate $\eta > 0$, ensures that for any machine $i \in [M]$, that acts at time-steps $\mathcal{T}_i$, the resulting sequences $\{\tilde{s}_{t,i}\}_{t \in \mathcal{T}_i}$ is $\frac{\rho_i^2}{2}$-zCDP, where: $\rho_i = 2S\sqrt{\sum_{t \in \mathcal{T}_i}\frac{1}{\sigma_{t,i}^2}}$.*

*Proof Sketch.* First, assume that $\mathcal{S}_i$ and $\mathcal{S}_i'$ are neighboring datasets, meaning that there exists only a single time-step $\tau^* \in \mathcal{T}_i$ where they differ, i.e. that $z_{\tau^*,i} \ne z'_{\tau^*,i}$.

We then use the post-processing property (Lemma A.4) to say that the privacy of $\{\tilde{s}_{t,i}\}_{t \in \mathcal{T}_i}$ is equal to the pri-

**Algorithm 1** DP-$\mu^2$-FL with Partial-Participation

---

**Inputs:** #iterations $T$, #TotalMachines $M$, #Machines per step $m$, initial point $x_0$, learning rate $\eta > 0$, importance weights $\{\alpha_t > 0\}$, noise distributions $\{P_{t,i} = \mathcal{N}(0, I\sigma_{t,i}^2)\}$, datasets $\{\mathcal{S}_i = \{z_{1,i}, \ldots, z_{T,i}\}\}$
**Initialize:** set $w_1 = x_1 = x_0$, $\tilde{q}_0 = 0$, and $Y_i = 0$, $\forall i$
**for** $t = 1, \ldots, T$ **do**
    Choose $\mathcal{M}_t \subseteq M$ with $|\mathcal{M}_t| = m$
    **for** every Machine $i \in \mathcal{M}_t$ **do**
        **Actions of Machine** $i$:
        Retrieve $z_{t,i}$ from $\mathcal{S}_i$, compute $g_{t,i} = \nabla f(x_t; z_{t,i})$,
        and $\tilde{g}_{t-1,i} = \nabla f(x_{t-1}; z_{t,i})$
        Update $s_{t,i} = \alpha_t g_{t,i} - \alpha_{t-1} \tilde{g}_{t-1,i}$
        Draw $y_{t,i} \sim \mathcal{N}(0, I\sigma_{t,i}^2)$
        Update $\tilde{s}_{t,i} = s_{t,i} + y_{t,i} - Y_i$
        Update $Y_i = y_{t,i}$ {saving the last noise}
    **end for**
    **Actions of Server:**
    Aggregate $\tilde{s}_t = \frac{1}{m} \sum_{i \in \mathcal{M}_t} \tilde{s}_{t,i}$
    Update $\tilde{q}_t = \tilde{q}_{t-1} + \tilde{s}_t$
    Update $w_{t+1} = \Pi_{\mathcal{K}}(w_t - \eta \tilde{q}_t)$
    Update $x_{t+1} = \left(1 - \frac{\alpha_{t+1}}{\alpha_{1:t+1}}\right) x_t + \frac{\alpha_{t+1}}{\alpha_{1:t+1}} w_{t+1}$
**end for**
**Output:** $x_T$

---

vacy of $\left\{\sum_{\tau \leq t \& \tau \in \mathcal{T}_i} \tilde{s}_{\tau,i}\right\}_{t \in \mathcal{T}_i}$. Using the composition rule (Lemma A.3), we bound the privacy of them with the sum of the privacy of each individual member. Now, recalling that the noises cancel each other and thus:

$$\sum_{\tau \leq t \& \tau \in \mathcal{T}_i} \tilde{s}_{\tau,i} = \sum_{\tau \leq t \& \tau \in \mathcal{T}_i} s_{\tau,i} + y_{t,i}, \quad \forall t \in \mathcal{T}_i$$

We may use Lemma 2.3 and obtain that $\sum_{\tau \leq t \& \tau \in \mathcal{T}_i} \tilde{s}_{\tau,i}$ is $\frac{\Delta_{t,i}^2}{2\sigma_{t,i}^2}$-zCDP. Using the bound $\|s_{\tau,i}\| \leq S$, which holds for any $\tau \in \mathcal{T}_i$ due to Lemma 4.2, we show that $\Delta_{t,i} \leq 2S$. Thus, we are $\frac{2S^2}{\sigma_{t,i}^2}$-zCDP. Using the above together, we get that $\{\tilde{s}_{t,i}\}_{t \in \mathcal{T}_i}$ is $2S^2 \sum_{t \in \mathcal{T}_i} \frac{1}{\sigma_{t,i}^2}$-zCDP. $\qquad \square$

### 5.2. Convergence Guarantees

Guarantees of Algorithm 1 (proof in Appendix E.2):

**Theorem 5.2.** *Let $\mathcal{K} \subset \mathbb{R}^d$ be a convex set of diameter $D$ and $\{f_i(\cdot; z)\}_{i \in [M], z \in \mathcal{Z}_i}$ be a family of $G$-Lipschitz and $L$-smooth functions over $\mathcal{K}$, with $\sigma, \xi \in [0, G], \sigma_L, \xi_L \in [0, L]$, and let $\{\mathcal{M}_t\}_t$ be subsets of $[M]$ of size $m$, define $G^* := \nabla f(x^*)$, where $x^* = \arg\min_{x \in \mathcal{K}} f(x)$, and $S := G + 2LD, \tilde{\sigma} := \sigma + 2\sigma_L D, \tilde{\xi} := \xi + 2\xi_L D$, moreover let $T \in \mathbb{N}, \rho > 0$.*

*Then upon invoking Algorithm 1 with $\alpha_t = t$,*

$\eta = \min\left\{\frac{\rho D m}{2ST\sqrt{2Md(1+\log T)}}, \frac{1}{8LT}\right\}$, and $\sigma_{t,i}^2 = \frac{4S^2(1+\log T)}{\rho^2} N_{t,i}$, with $N_{t,i}$ being the number of time steps that machine $i$ participated up to time step $t$, and for any datasets $\{\mathcal{S}_i \in \mathcal{Z}_i^T\}_{i \in [M]}$, then Algorithm 1 satisfies $\frac{\rho^2}{2}$-zCDP w.r.t gradient estimate correction sequences that each machine produces, i.e. $\{\tilde{s}_{t,i}\}_{i \in [M], t \in \mathcal{T}_i}$.

*Furthermore, if $\mathcal{S}_i$ consists of i.i.d. samples from a distribution $\mathcal{D}_i$ for all $i \in [M]$, and $\mathcal{M}_t$ are also chosen uniformly in an i.i.d manner, then Algorithm 1 guarantees:*

$$\mathcal{R}_T := \mathbb{E}[f(x_T)] - \min_{x \in \mathcal{K}} f(x) \leq$$

$$4D\left(\frac{G^* + 4LD}{T} + \frac{\sqrt{\tilde{\sigma}^2 + \tilde{\xi}^2}}{\sqrt{mT}} + \frac{2S\sqrt{2Md(1+\log T)}}{\rho m T}\right)$$

*Proof Sketch.* The privacy guarantees follow directly from Theorem 5.1, and our choice of $\sigma_{t,i}^2$:

$$2S^2 \sum_{t \in \mathcal{T}_i} \frac{1}{\sigma_{t,i}^2} = 2S^2 \sum_{t \in \mathcal{T}_i} \frac{\rho^2}{4S^2(1+\log T) N_{t,i}}$$

$$= \frac{\rho^2}{2(1+\log T)} \sum_{t \in \mathcal{T}_i} \frac{1}{N_{t,i}} = \frac{\rho^2}{2(1+\log T)} \sum_{k=1}^{|\mathcal{T}_i|} \frac{1}{k}$$

$$\leq \frac{\rho^2}{2(1+\log T)}(1 + \log|\mathcal{T}_i|) \leq \frac{\rho^2}{2}$$

Regrading convergence, in the spirit of $\mu^2$-SGD analysis (Levy, 2023; Reshef & Levy, 2024), we bound the excess loss using the anytime theorem (Theorem A.1), rewrite the expression to get to the form of Lemma A.5 and use it to bound, and separate the terms of $\varepsilon_t$ and $Y_t$. We already bounded $\varepsilon_t$ in Lemma 4.3, though we bound $\frac{M-m}{M-1} \leq 1$, but the bound on the parts of $Y_t$ is harder, since the sequence $\{Y_t\}_{t \in T}$ is not independent of each other, as it was in (Reshef & Levy, 2024), since in each round different machine participate. We get:

$$\sum_{\tau=1}^t \mathbb{E}\langle Y_\tau, x^* - w_{\tau+1}\rangle \leq \frac{1}{4\eta} \sum_{r=1}^t \mathbb{E}\|w_r - w_{r+1}\|^2$$

$$+ \frac{\eta}{m^2 p(2-p)} \sum_{i=1}^M \sum_{s=1}^t \mathbb{E}\|y_{s,i}\|^2$$

Where $p = \frac{m}{M}$ is the probability that machine $i$ participate at time-step $t$ for all machines and time-steps. We then use our chosen value for $\sigma_{s,i}^2$ to bound $y_{s,i}$. We then input all the bounds, bound the gradient using the excess loss with Lemma A.6, to get a bound of the excess loss using the previous excess losses. Using Lemma A.7 we get the final bound on the excess loss. By inputting our chosen $\eta$, that minimize this expression, we get our bound. $\qquad \square$



*Table 1.* Privacy Level Comparison

| | Our Work | | Noisy SGD | | Other Work | |
|---|---|---|---|---|---|---|
| $\rho$ | Accuracy | Time | Accuracy | Time | Accuracy | Time |
| 4 | 53.8% | 13 sec | 45.1% | 9 sec | 47.6% | 64 sec |
| 8 | 63.7% | 13 sec | 58.9% | 9 sec | 63.3% | 282 sec |
| 12 | 66.5% | 13 sec | 63.7% | 9 sec | 66.7% | 730 sec |

*Table 2.* Participating Machines Comparison

| | Our Work | | Noisy SGD | | Other Work | |
|---|---|---|---|---|---|---|
| $m$ | Accuracy | Time | Accuracy | Time | Accuracy | Time |
| 20 | 60.8% | 13 sec | 54.9% | 9 sec | 59.7% | 114 sec |
| 50 | 63.7% | 13 sec | 58.9% | 9 sec | 63.3% | 282 sec |
| 80 | 63.8% | 13 sec | 57.0% | 9 sec | 65.8% | 452 sec |



Since Algorithm 1 uses a total of $n = mT$ samples in the learning process, our rate translates to $O\left(\frac{1}{\sqrt{n}} + \frac{\sqrt{M}d}{\epsilon n}\right)$ which matches the lower bound for the untrusted server case. Moreover, our approach performs two gradient computations per sample, and its computational complexity is therefore linear in $n$, which matches standard methods.

### 5.3. Experiments

We ran Algorithm 1 on MNIST using a logistic regression. The parameters are $G = \sqrt{2 \cdot 785} = 39.6, L = 785/2 = 392.5, D = 0.1$, which brings us $S = 118.1$. Our model has $d = 10 \cdot 785 = 7850$ parameters. We compared our algorithm (called "Our Work") to SGD with noise, inspired by (Abadi et al., 2016) (called "Noisy SGD"), and to the other work (Lowy & Razaviyayn, 2023) (called "Other Work"). We kept the same parameter in all 3 algorithms to the best of our abilities, and in all tests the total data samples used across all machines is $n = 60,000$. Note that in Our Work and Noisy SGD, we only do a single-pass over the data, and each data sample is only used once, while in the Other Work the same samples are reused, as is done in their algorithm. This fact and the noise added as part of privacy are part of the reason why the accuracy is bellow 70% in all experiments. We compare both the test accuracy and running time.

For our first experiment, we fix $m = 50, M = 100$, and compare various values of $\rho$. We show our results in Table 1.

We can see that by increasing $\rho$, the accuracy of all algorithms increase. That makes sense, since the higher the privacy level, the less noise we need. The running time doesn't change in Our Work or Noisy SGD, because we make the same number of computations, just with different noise, but in the Other Work, they use more iterations with a higher privacy level, to get better results when allowed, so the running time drastically increases. Comparing between the algorithms, the Other Work needs large privacy level to truly shine, overtaking Our Work in the highest privacy level, but getting results closer to Noisy SGD in the lowest, albeit with a much longer running time, with our algorithm being just slightly longer than Noisy SGD, with better accuracy.

In a second experiment, we fix $\rho = 8, M = 100$, and compare various values of $m$. We show our results in Table 2.

We can see that by increasing $m$, we increase the accuracy, except in Noisy SGD, where the higher value is worse than the middle one. That is because in Our Work and the Other Work, the less partial the participation is, the less we lose, but it doesn't matter for Noisy SGD. In Noisy SGD, $m$ can be seen as the batch size, which is inverse to the number of time-steps, which was probably too small in this case. The change in accuracy is smaller than the one in Table 1, which implies that the privacy is more important, as can be seen in our bound in Theorem 5.2. The running time is unchanged in Our Work and Noisy SGD, since the total number of samples used is the same, but is increasing in the Other Work with $m$, since they make more iterations when $m$ is larger. Comparing between the algorithms, the Other Work gets the best accuracy in the largest $m$, but does so with a much longer running time. Our Work still has slightly longer running time than Noisy SGD with better accuracy.

In conclusion, like it was shown from the theoretical bounds, Our Work is as good as the Other Work in terms of accuracy, but with much faster running time, while getting only a slightly longer running time than Noisy SGD but with a better accuracy.

## 6. Conclusion

We enhanced DP-$\mu^2$-FL to operate in the partial-participation setting with an untrusted server by introducing an innovative noise cancellation technique that preserves privacy while minimizing overall noise. We demonstrated that this approach achieves optimal excess loss with linear computational complexity.

### Impact Statement

This paper presents work whose goal is to advance the field of Machine Learning, and especially the aspect of Privacy. There are many potential societal consequences of our work, none which we feel must be specifically highlighted here.

### Acknowledgement

This research was partially supported by Israel PBC- VATAT, by the Technion Artificial Intelligent Hub (Tech.AI), and by the Israel Science Foundation (grant No. 3109/24).

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

# A. Additional Theorems and Lemmas

Here we provide additional theorems and lemmas that are used for our proofs. These were also used in (Reshef & Levy, 2024).

**Theorem A.1** ((Cutkosky, 2019)). *Let $f : \mathcal{K} \to \mathbb{R}$ be a convex function. Also let $\{\alpha_t > 0\}$ and $\{w_t \in \mathcal{K}\}$. Let $\{x_t\}$ be the $\{\alpha_\tau\}_{\tau=1}^t$ weighted average of $\{w_\tau\}_{\tau=1}^t$, meaning: $x_t = \frac{1}{\alpha_{1:t}} \sum_{\tau=1}^t \alpha_\tau w_\tau$. Then the following holds for all $t \geq 1, x \in \mathcal{K}$:*

$$\alpha_{1:t}(f(x_t) - f(x)) \leq \sum_{\tau=1}^t \alpha_\tau \langle \nabla f(x_\tau), w_\tau - x \rangle$$

Note that the above theorem holds generally for any sequences of iterates $\{w_t\}_t$ with weighted averages $\{x_t\}_t$, and as a private case it holds for the sequences generated by Anytime-SGD. Concretely, the theorem implies that the excess loss of the weighted average $x_t$ can be related to the weighted regret $\sum_{\tau=1}^t \alpha_\tau \langle \nabla f(x_\tau), w_\tau - x \rangle$.

*Proof of Theorem A.1.* Proof by induction.
**Induction basis:** $t = 1$

$$\alpha_1(f(x_1) - f(x)) \leq \alpha_1 \langle \nabla f(x_1), x_1 - x \rangle = \alpha_1 \langle \nabla f(x_1), w_1 - x \rangle$$

The inequality is from convexity of $f$, and the equality is because $x_1 = w_1$.
**Induction assumption:** for some $t \geq 1$:

$$\alpha_{1:t}(f(x_t) - f(x)) \leq \sum_{\tau=1}^t \alpha_\tau \langle \nabla f(x_\tau), w_\tau - x \rangle$$

**Induction step:** proof for $t + 1$:

$$\begin{aligned}
\alpha_{1:t+1}(f(x_{t+1}) - f(x)) =& \alpha_{1:t}(f(x_{t+1}) - f(x_t) + f(x_t) - f(x)) + \alpha_{t+1}(f(x_{t+1}) - f(x)) \\
\leq& \sum_{\tau=1}^t \alpha_\tau \langle \nabla f(x_\tau), w_\tau - x \rangle + \alpha_{1:t}(f(x_{t+1}) - f(x_t)) + \alpha_{t+1}(f(x_{t+1}) - f(x)) \\
\leq& \sum_{\tau=1}^t \alpha_\tau \langle \nabla f(x_\tau), w_\tau - x \rangle + \langle \nabla f(x_{t+1}), \alpha_{1:t}(x_{t+1} - x_t) + \alpha_{t+1}(x_{t+1} - x) \rangle \\
=& \sum_{\tau=1}^t \alpha_\tau \langle \nabla f(x_\tau), w_\tau - x \rangle + \alpha_{t+1} \langle \nabla f(x_{t+1}), w_{t+1} - x \rangle = \sum_{\tau=1}^{t+1} \alpha_\tau \langle \nabla f(x_\tau), w_\tau - x \rangle
\end{aligned}$$

In the first equality we rearranged the terms, and added and subtracted the same thing, we then used the induction assumption, then used convexity of $f$ on both pairs and added them together, and finally we used the update rule of $x_{t+1}$: $\alpha_{1:t+1}x_{t+1} = \alpha_{1:t}x_t + \alpha_{t+1}w_{t+1}$, and added the last member of the sum to get our desired result. $\square$

**Lemma A.2.** *Let $\{Z_t\}$ be a Martingale difference sequence w.r.t a Filtration $\{\mathcal{F}_t\}_t$, i.e. $\mathbb{E}[Z_t|\mathcal{F}_{t-1}] = 0$, then:*

$$\mathbb{E} \left\| \sum_{\tau=1}^t Z_\tau \right\|^2 = \sum_{\tau=1}^t \mathbb{E} \|Z_\tau\|^2$$

*Proof of Lemma A.2.* Proof by induction.
**Induction basis:** $t = 1$

$$\mathbb{E} \left\| \sum_{\tau=1}^1 Z_\tau \right\|^2 = \mathbb{E} \|Z_1\|^2 = \sum_{\tau=1}^1 \mathbb{E} \|Z_\tau\|^2$$

**Induction assumption:** for some $t \geq 1$:

$$\mathbb{E} \left\| \sum_{\tau=1}^{t} Z_\tau \right\|^2 = \sum_{\tau=1}^{t} \mathbb{E} \|Z_\tau\|^2$$

**Induction step:** proof for $t+1$:

$$\mathbb{E} \left\| \sum_{\tau=1}^{t+1} Z_\tau \right\|^2 = \mathbb{E} \left\| \sum_{\tau=1}^{t+1} Z_\tau \right\|^2 + 2\mathbb{E} \left\langle \sum_{\tau=1}^{t} Z_\tau, Z_{t+1} \right\rangle + \mathbb{E} \|Z_{t+1}\|^2 = \sum_{\tau=1}^{t} \mathbb{E} \|Z_\tau\|^2 + \mathbb{E} \|Z_{t+1}\|^2 = \sum_{\tau=1}^{t+1} \mathbb{E} \|Z_\tau\|^2$$

The first equality is square rules, then we used the induction assumption and the fact that $\mathbb{E}[Z_{t+1}|Z_1, \ldots, Z_t] = 0$, and finally we added the last term into the sum. $\square$

**Lemma A.3** ((Mironov, 2017; Bun & Steinke, 2016)). *If $\mathcal{A}_1, \ldots, \mathcal{A}_k$ are randomized algorithms satisfying $\rho_1$-zCDP, ..., $\rho_k$-zCDP, respectively, then their composition $(\mathcal{A}_1(\mathcal{S}) \ldots, \mathcal{A}_k(\mathcal{S}))$ is $(\rho_1 + \ldots, +\rho_k)$-zCDP. Moreover, the i'th algorithm $\mathcal{A}_i$, can be chosen on the basis of the outputs of the previous algorithms $\mathcal{A}_1, \ldots, \mathcal{A}_{i-1}$.*

*Proof of Lemma A.3.* Proof by induction.
**Induction basis:** $k = 1$

$$\mathbb{D}_\alpha \left( \mathcal{A}_1(\mathcal{S}) \| \mathcal{A}_1(\mathcal{S}') \right) \leq \alpha \rho_1$$

Because $\mathcal{A}_1$ is $\rho_1$-zCDP.
**Induction assumption:** for some $k \geq 1$:

$$\mathbb{D}_\alpha \left( \{\mathcal{A}_i(\mathcal{S})\}_{i=1}^{k} \| \{\mathcal{A}_i(\mathcal{S}')\}_{i=1}^{k} \right) \leq \alpha \sum_{i=1}^{k} \rho_i$$

**Induction step:** proof for $k+1$

$$\mathbb{D}_\alpha \left( \{\mathcal{A}_i(\mathcal{S})\}_{i=1}^{k+1} \| \{\mathcal{A}_i(\mathcal{S}')\}_{i=1}^{k+1} \right) = \frac{1}{\alpha-1} \log \left( \mathbb{E}_{\mathcal{A}_i \sim \mathcal{A}_i(\mathcal{S})} \left[ \left( \frac{\mathbb{P}\left\{ \{\mathcal{A}_i(\mathcal{S})\}_{i=1}^{k+1} \right\}}{\mathbb{P}\left\{ \{\mathcal{A}_i(\mathcal{S}')\}_{i=1}^{k+1} \right\}} \right)^{\alpha-1} \right] \right)$$

$$= \frac{1}{\alpha-1} \log \left( \mathbb{E}_{\mathcal{A}_i \sim \mathcal{A}_i(\mathcal{S})} \left[ \left( \frac{\mathbb{P}\left\{ \mathcal{A}_{k+1}(\mathcal{S}) | \{\mathcal{A}_i\}_{i=1}^{k} \right\} \mathbb{P}\left\{ \{\mathcal{A}_i(\mathcal{S})\}_{i=1}^{k} \right\}}{\mathbb{P}\left\{ \mathcal{A}_{k+1}(\mathcal{S}') | \{\mathcal{A}_i\}_{i=1}^{k} \right\} \mathbb{P}\left\{ \{\mathcal{A}_i(\mathcal{S}')\}_{i=1}^{k} \right\}} \right)^{\alpha-1} \right] \right)$$

$$= \frac{1}{\alpha-1} \log \left( \mathbb{E}_{\mathcal{A}_i \sim \mathcal{A}_i(\mathcal{S})} \left[ \left( \frac{\mathbb{P}\left\{ \mathcal{A}_{k+1}(\mathcal{S}) | \{\mathcal{A}_i\}_{i=1}^{k} \right\}}{\mathbb{P}\left\{ \mathcal{A}_{k+1}(\mathcal{S}') | \{\mathcal{A}_i\}_{i=1}^{k} \right\}} \right)^{\alpha-1} \right] \right)$$

$$+ \frac{1}{\alpha-1} \log \left( \mathbb{E}_{\mathcal{A}_i \sim \mathcal{A}_i(\mathcal{S})} \left[ \left( \frac{\mathbb{P}\left\{ \{\mathcal{A}_i(\mathcal{S})\}_{i=1}^{k} \right\}}{\mathbb{P}\left\{ \{\mathcal{A}_i(\mathcal{S}')\}_{i=1}^{k} \right\}} \right)^{\alpha-1} \right] \right)$$

$$= \mathbb{D}_\alpha \left( \mathcal{A}_{k+1}(\mathcal{S}) \| \mathcal{A}_{k+1}(\mathcal{S}') | \{\mathcal{A}_i\}_{i=1}^{k} \right) + \mathbb{D}_\alpha \left( \{\mathcal{A}_i(\mathcal{S})\}_{i=1}^{k} \| \{\mathcal{A}_i(\mathcal{S}')\}_{i=1}^{k} \right)$$

$$\leq \alpha \rho_{k+1} + \alpha \sum_{i=1}^{k} \rho_i = \alpha \sum_{i=1}^{k+1} \rho_i$$

The writing $\mathcal{A}_i \sim \mathcal{A}_i(\mathcal{S})$ means that the output of $\mathcal{A}_i$ is distributed in the case that the dataset is $\mathcal{S}$. The first equation is the definition of the Rényi divergence, next we use conditional probability rules, then we use the fact that the output of $\mathcal{A}_{k+1}$ conditioned on the outputs of the previous algorithms is independent of the outputs of these previous algorithms, and let the product out of the log as a summery. We then notice that each of the two members are divergences themselves, and finally, we invoke both the the fact that $\mathcal{A}_{k+1}$ is $\rho_{k+1}$-zCDP, and the induction assumption, and add them together to the same sum. $\square$

**Lemma A.4** (Post Processing Lemma (van Erven & Harremoës, 2014))**.** *Let $X, Y$ be random variables and $\mathcal{A}$ be a randomized or deterministic algorithm. Then for all $\alpha \geq 1$:*

$$\mathbb{D}_\alpha \left(\mathcal{A}(X) \| \mathcal{A}(Y)\right) \leq \mathbb{D}_\alpha \left(X \| Y\right)$$

Note that we get equality if $\mathcal{A}$ is an invertible function.

**Lemma A.5.** *Let $\eta > 0$, and $\mathcal{K} \subset \mathbb{R}^d$ be a convex domain of bounded diameter $D$, also let $\{\tilde{q}_t \in \mathbb{R}^d\}_{t=1}^T$ be a sequence of arbitrary vectors. Then for any starting point $w_1 \in \mathbb{R}^d$, and an update rule $w_{t+1} = \Pi_\mathcal{K} \left(w_t - \eta\tilde{q}_t\right)$ , $\forall t \geq 1$, the following holds $\forall x \in \mathcal{K}$:*

$$\sum_{\tau=1}^t \langle \tilde{q}_\tau, w_{\tau+1} - x \rangle \leq \frac{D^2}{2\eta} - \frac{1}{2\eta} \sum_{\tau=1}^t \|w_\tau - w_{\tau+1}\|^2$$

*Proof of Lemma A.5.* The update rule $w_{t+1} = \Pi_\mathcal{K} \left(w_t - \eta\tilde{q}_t\right)$ can be re-written as a convex optimization problem over $\mathcal{K}$:

$$w_{t+1} = \Pi_\mathcal{K} \left(w_t - \eta\tilde{q}_t\right) = \arg\min_{x\in\mathcal{K}} \left\{ \|w_t - \eta\tilde{q}_t - x\|^2 \right\} = \arg\min_{x\in\mathcal{K}} \left\{ \langle \tilde{q}_t, x - w_t \rangle + \frac{1}{2\eta} \|x - w_t\|^2 \right\}$$

The first equality is our update definition, the second is by the definition of the projection operator, and then we rewrite it in a way that does not affect the minimum point.

Now, since $w_{t+1}$ is the minimal point of the above convex problem, then from optimality conditions we obtain:

$$\left\langle \tilde{q}_t + \frac{1}{\eta}(w_{t+1} - w_t), x - w_{t+1} \right\rangle \geq 0, \quad \forall x \in \mathcal{K}$$

Re-arranging the above, we get that:

$$\langle \tilde{q}_t, w_{t+1} - x \rangle \leq \frac{1}{\eta} \langle w_t - w_{t+1}, w_{t+1} - x \rangle = \frac{1}{2\eta} \|w_t - x\|^2 - \frac{1}{2\eta} \|w_{t+1} - x\|^2 - \frac{1}{2\eta} \|w_t - w_{t+1}\|^2$$

Where the equality is an algebraic manipulation. After summing over $t$ we get:

$$\sum_{\tau=1}^t \langle \tilde{q}_\tau, w_{\tau+1} - x \rangle \leq \frac{1}{2\eta} \sum_{\tau=1}^t \left( \|w_\tau - x\|^2 - \|w_{\tau+1} - x\|^2 - \|w_\tau - w_{\tau+1}\|^2 \right)$$

$$= \frac{\|w_1 - x\|^2 - \|w_{t+1} - x\|^2}{2\eta} - \frac{1}{2\eta} \sum_{\tau=1}^t \|w_\tau - w_{\tau+1}\|^2 \leq \frac{D^2}{2\eta} - \frac{1}{2\eta} \sum_{\tau=1}^t \|w_\tau - w_{\tau+1}\|^2$$

Where the second line is due to splitting the sum into two sums, and using the fact that the first one is a telescopic sum, and lastly, we use the diameter of $\mathcal{K}$. This establishes the lemma. $\qquad\square$

**Lemma A.6.** *If $f : \mathcal{K} \to \mathbb{R}$ is convex and $L$-smooth, and $x^* = \arg\min_{x\in\mathcal{K}} \{f(x)\}$, then $\forall x \in \mathbb{R}^d$:*

$$\|\nabla f(x) - \nabla f(x^*)\|^2 \leq 2L(f(x) - f(x^*))$$

*Proof of Lemma A.6.* Let us define a new function:

$$h(x) = f(x) - f(x^*) - \langle \nabla f(x^*), x - x^* \rangle$$

Since $f$ is convex and $L$-smooth, we know that:

$$0 \leq h(x) \leq \frac{L}{2} \|x - x^*\|^2$$

The gradient of this function is:

$$\nabla h(x) = \nabla f(x) - \nabla f(x^*)$$

We can see that $h(x^*) = 0, \nabla h(x^*) = 0$, and that $x^*$ is the global minimum. The function $h$ is also convex and $L$-smooth, since the gradient is the same as $f$ up to a constant translation. We will add to the domain of $h$ to include all $\mathbb{R}^d$, while still being convex and $L$-smooth. Since $h$ is convex then:

$$h(y) \geq h(x^*) + \langle \nabla h(x^*), y - x^* \rangle = 0, \quad \forall y \in \mathbb{R}^d$$

It is true even for points outside of the original domain, meaning that $x^*$ remains the global minimum even after this. For a smooth function, $\forall x, y \in \mathbb{R}^d$:

$$h(y) \leq h(x) + \langle \nabla h(x), y - x \rangle + \frac{L}{2} \|y - x\|^2$$

By picking $y = x - \frac{1}{L} \nabla h(x)$, we get:

$$h(x) - h(y) \geq \frac{1}{2L} \|\nabla h(x)\|^2$$

Rearranging, we get:

$$\|\nabla h(x)\|^2 \leq 2L(h(x) - h(y)) \leq 2L \cdot h(x)$$

By using $x \in \mathcal{K}$ we get:

$$\|\nabla f(x) - \nabla f(x^*)\|^2 \leq 2L(f(x) - f(x^*) - \langle \nabla f(x^*), x - x^* \rangle$$

Since $x^*$ is the minimum point of $f$ then:

$$\langle \nabla f(x^*), x - x^* \rangle \geq 0, \quad \forall x \in \mathcal{K}$$

Thus we get that:

$$\|\nabla f(x) - \nabla f(x^*)\|^2 \leq 2L(f(x) - f(x^*))$$

$\square$

**Lemma A.7.** *If* $\mathcal{A}_t \leq \frac{1}{2T} \sum_{\tau=1}^{T} \mathcal{A}_\tau + \mathcal{B}, \forall t \in [T]$, *then* $\mathcal{A}_t \leq 2\mathcal{B}, \forall t \in [T]$.

*Proof of Lemma A.7.* Let's sum the inequalities:

$$\sum_{t=1}^{T} \mathcal{A}_t \leq \sum_{t=1}^{T} \left( \frac{1}{2T} \sum_{\tau=1}^{T} \mathcal{A}_\tau + \mathcal{B} \right) = \frac{1}{2} \sum_{\tau=1}^{T} \mathcal{A}_\tau + T\mathcal{B}$$

The inequality is from the assumption, and the equality is because we sum constant values. If we rearrange this we get:

$$\sum_{\tau=1}^{T} \mathcal{A}_\tau \leq 2T\mathcal{B}$$

And then:

$$\mathcal{A}_t \leq \frac{1}{2T} \sum_{\tau=1}^{T} \mathcal{A}_\tau + \mathcal{B} \leq \mathcal{B} + \mathcal{B} = 2\mathcal{B}$$

$\square$

# B. Proofs of Section 2

## B.1. Proof of Equations (4) to (7)

Both claims use the same principle:

$$\mathbb{E}\left\|X - \mathbb{E}\left[X\right]\right\|^2 \le \mathbb{E}\left\|X\right\|^2$$

And so:

$$\mathbb{E}\left\|\nabla f_i(x; z) - \nabla f_i(x)\right\|^2 \le \mathbb{E}\left\|\nabla f_i(x; z)\right\|^2 \le G^2$$
$$\mathbb{E}\left\|(\nabla f_i(x; z) - \nabla f_i(x)) - (\nabla f_i(y; z) - \nabla f_i(y))\right\|^2 \le \mathbb{E}\left\|\nabla f_i(x; z) - \nabla f_i(y; z)\right\|^2 \le L^2\left\|x - y\right\|^2$$

With the heterogeneity, $\nabla f(x)$ is the empirical mean of $\nabla f_i(x)$, so the same rule applies:

$$\frac{1}{M}\sum_{i=1}^{M}\left\|\nabla f_i(x) - \nabla f(x)\right\|^2 \le \mathbb{E}\left\|\nabla f_i(x)\right\|^2 \le G^2$$

$$\frac{1}{M}\sum_{i=1}^{M}\left\|(\nabla f_i(x) - \nabla f(x)) - (\nabla f_i(y) - \nabla f(y))\right\|^2 \le \mathbb{E}\left\|\nabla f_i(x) - \nabla f_i(y)\right\|^2 \le L^2\left\|x - y\right\|^2$$

Thus $\sigma, \xi \le G$ and $\sigma_L, \xi_L \le L$.

## B.2. Proof of Lemma 2.1

The first part will be a combination of the bounded variance and the bounded heterogeneity. At first, we will assume that $\mathcal{M}$ is fixed:

$$\mathbb{E}\left\|g(x) - \nabla f(x)\right\|^2 = \mathbb{E}\left\|\frac{1}{m}\sum_{i\in\mathcal{M}}\nabla f_i(x; z_i) - \nabla f(x)\right\|^2$$

$$= \left\|\frac{1}{m}\sum_{i\in\mathcal{M}}\nabla f_i(x) - \nabla f(x)\right\|^2 + \mathbb{E}\left\|\frac{1}{m}\sum_{i\in\mathcal{M}}\nabla f_i(x; z_i) - \nabla f_i(x)\right\|^2$$

$$= \left\|\frac{1}{m}\sum_{i\in\mathcal{M}}\nabla f_i(x) - \nabla f(x)\right\|^2 + \frac{1}{m^2}\sum_{i\in\mathcal{M}}\mathbb{E}\left\|\nabla f_i(x; z_i) - \nabla f_i(x)\right\|^2$$

$$\le \left\|\frac{1}{m}\sum_{i\in\mathcal{M}}\nabla f_i(x) - \nabla f(x)\right\|^2 + \frac{1}{m^2}\sum_{i\in\mathcal{M}}\sigma^2$$

$$= \left\|\frac{1}{m}\sum_{i\in\mathcal{M}}\nabla f_i(x) - \nabla f(x)\right\|^2 + \frac{\sigma^2}{m}$$

At first we use the definition of $g(x)$, then we use the fact that $\mathbb{E}\left\|X\right\|^2 = \left\|\mathbb{E}X\right\|^2 + \mathbb{E}\left\|X - \mathbb{E}X\right\|^2$ and $\mathbb{E}\left[\nabla f_i(x; z_i)\right] = \nabla f_i(x)$. Then we use the fact that the variance of a sum equal to the sum of the variances, and then use the bounded variance. Next we need to bound the first term:

$$\left\|\frac{1}{m}\sum_{i\in\mathcal{M}}\nabla f_i(x) - \nabla f(x)\right\|^2 = \frac{1}{m^2}\left\|\sum_{i=1}^{M}\mathbb{I}\left\{i\in\mathcal{M}\right\}(\nabla f_i(x) - \nabla f(x))\right\|^2$$

$$= \frac{1}{m^2}\sum_{i=1}^{M}\sum_{j=1}^{M}\mathbb{I}\left\{i\in\mathcal{M}\right\}\mathbb{I}\left\{j\in\mathcal{M}\right\}\left\langle\nabla f_i(x) - \nabla f(x), \nabla f_j(x) - \nabla f(x)\right\rangle$$

Where we rewrote the sum using the indicator function, and then open the square into two sums. Now we will add an expectation on the randomness of $\mathcal{M}$. Note that the only random variables here are the indicators. The properties of the

indicators, given that $\mathcal{M}$ is chosen uniformly:

$$\mathbb{E}\left[\mathbb{I}\{i \in \mathcal{M}\}\mathbb{I}\{j \in \mathcal{M}\}\right] = \mathbb{P}\{i, j \in \mathcal{M}\} = \begin{cases} \frac{m}{M} & i = j \\ \frac{m(m-1)}{M(M-1)} & i \neq j \end{cases}$$

$$= \frac{m(m-1)}{M(M-1)} + \left(\frac{m}{M} - \frac{m(m-1)}{M(M-1)}\right)\mathbb{I}\{i = j\} = \frac{m(m-1)}{M(M-1)} + \frac{m(M-m)}{M(M-1)}\mathbb{I}\{i = j\}$$

Where at the end, we wrote it using an indicator, for the case they are equal. Returning to the bound, we get:

$$\mathbb{E}\left\|\frac{1}{m}\sum_{i \in \mathcal{M}}\nabla f_i(x) - \nabla f(x)\right\|^2$$

$$= \frac{1}{m^2}\sum_{i=1}^{M}\sum_{j=1}^{M}\left(\frac{m(m-1)}{M(M-1)} + \frac{m(M-m)}{M(M-1)}\mathbb{I}\{i = j\}\right)\langle\nabla f_i(x) - \nabla f(x), \nabla f_j(x) - \nabla f(x)\rangle$$

$$= \frac{m-1}{mM(M-1)}\sum_{i=1}^{M}\sum_{j=1}^{M}\langle\nabla f_i(x) - \nabla f(x), \nabla f_j(x) - \nabla f(x)\rangle + \frac{M-m}{mM(M-1)}\sum_{i=1}^{M}\|\nabla f_i(x) - \nabla f(x)\|^2$$

$$= \frac{M-m}{m(M-1)}\frac{1}{M}\sum_{i=1}^{M}\|\nabla f_i(x) - \nabla f(x)\|^2 \leq \frac{M-m}{M-1}\frac{\xi^2}{m}$$

Where at first we inputted the expectation of the indicators, then separated the sum into two, then we see that the first sum is a multiplication of two independent sums, that are both equal to 0, since $\nabla f(x) = \frac{1}{M}\sum_{i=1}^{M}\nabla f_i(x)$, and finally we use the definition of $\xi$. In total we got that:

$$\mathbb{E}\|g(x) - \nabla f(x)\|^2 \leq \frac{1}{m}\left(\sigma^2 + \frac{M-m}{M-1}\xi^2\right)$$

For the second part, we will use similar steps, but with the bounded smoothness variance:

$$\mathbb{E}\|(g(x) - \nabla f(x)) - (g(y) - \nabla f(y))\|^2 = \mathbb{E}\left\|\frac{1}{m}\sum_{i \in \mathcal{M}}((\nabla f_i(x; z_i) - \nabla f(x)) - (\nabla f_i(y; z_i) - \nabla f(y)))\right\|^2$$

$$= \left\|\frac{1}{m}\sum_{i \in \mathcal{M}}((\nabla f_i(x) - \nabla f(x)) - (\nabla f_i(y) - \nabla f(y)))\right\|^2$$

$$+ \mathbb{E}\left\|\frac{1}{m}\sum_{i \in \mathcal{M}}((\nabla f_i(x; z_i) - \nabla f_i(x)) - (\nabla f_i(y; z_i) - \nabla f_i(y)))\right\|^2$$

Where we first used the definitions of $g(x), g(y)$ and the variance rule, the same steps as before. Also as before, we will bound the second term using the smoothness variance bound:

$$\mathbb{E}\left\|\frac{1}{m}\sum_{i \in \mathcal{M}}((\nabla f_i(x; z_i) - \nabla f_i(x)) - (\nabla f_i(y; z_i) - \nabla f_i(y)))\right\|^2$$

$$= \frac{1}{m^2}\sum_{i \in \mathcal{M}}\mathbb{E}\|((\nabla f_i(x; z_i) - \nabla f_i(x)) - (\nabla f_i(y; z_i) - \nabla f_i(y)))\|^2$$

$$\leq \frac{1}{m^2}\sum_{i \in \mathcal{M}}\sigma_L^2\|x - y\|^2 = \frac{\sigma_L^2}{m}\|x - y\|^2$$

The first term is the difficult one, so we will use indicators, as before. For simplicity of the writing, we will define $\Delta_i = (\nabla f_i(x) - \nabla f(x)) - (\nabla f_i(y) - \nabla f(y))$

$$\left\|\frac{1}{m}\sum_{i \in \mathcal{M}}\Delta_i\right\|^2 = \frac{1}{m^2}\left\|\sum_{i=1}^{M}\mathbb{I}\{i \in \mathcal{M}\}\Delta_i\right\|^2 = \frac{1}{m^2}\sum_{i=1}^{M}\sum_{j=1}^{M}\mathbb{I}\{i \in \mathcal{M}\}\mathbb{I}\{j \in \mathcal{M}\}\langle\Delta_i, \Delta_j\rangle$$

And just as before, we will use expectation over $\mathcal{M}$:

$$
\mathbb{E} \left\| \frac{1}{m} \sum_{i \in \mathcal{M}} \Delta_i \right\|^2 = \frac{1}{m^2} \sum_{i=1}^{M} \sum_{j=1}^{M} \left( \frac{m(m-1)}{M(M-1)} + \frac{m(M-m)}{M(M-1)} \mathbb{I}\left\{i=j\right\} \right) \langle \Delta_i, \Delta_j \rangle
$$

$$
= \frac{m-1}{mM(M-1)} \sum_{i=1}^{M} \sum_{j=1}^{M} \langle \Delta_i, \Delta_j \rangle + \frac{M-m}{mM(M-1)} \sum_{i=1}^{M} \|\Delta_i\|^2
$$

$$
= \frac{M-m}{m(M-1)} \frac{1}{M} \sum_{i=1}^{M} \|\Delta_i\|^2 \le \frac{M-m}{M-1} \frac{\xi_L^2}{m} \|x-y\|^2
$$

Where we used the same steps, including the fact that $\sum_{i=1}^{M} \Delta_i = 0$ and the definition of $\xi_L$. In total we got that:

$$
\mathbb{E} \left\| (g(x) - \nabla f(x)) - (g(y) - \nabla f(y)) \right\|^2 \le \frac{1}{m} \left( \sigma_L^2 + \frac{M-m}{M-1} \xi_L^2 \right) \|x-y\|^2
$$

## B.3. Proof of Lemma 2.3

Let us calculate $\mathbb{D}_\alpha (P\|Q)$ by definition.

$$
\mathbb{D}_\alpha (P\|Q) = \frac{1}{\alpha-1} \log \left( \mathbb{E}_{X \sim P} \left[ \left( \frac{P(X)}{Q(X)} \right)^{\alpha-1} \right] \right) = \frac{1}{\alpha-1} \log \left( \mathbb{E}_{X \sim P} \left[ e^{(\alpha-1)\frac{1}{2\sigma^2}\left( \|X-\mu-\Delta\|^2 - \|X-\mu\|^2 \right)} \right] \right)
$$

$$
= \frac{1}{\alpha-1} \log \left( \mathbb{E}_{X \sim P} \left[ e^{\frac{\alpha-1}{2\sigma^2}\left( \|\Delta\|^2 - 2\langle \Delta, X-\mu \rangle \right)} \right] \right) = \frac{1}{\alpha-1} \log \left( e^{\frac{\alpha-1}{2\sigma^2}\|\Delta\|^2 + \frac{1}{2}\sigma^2 \frac{(\alpha-1)^2}{4\sigma^4} 4\|\Delta\|^2} \right)
$$

$$
= \frac{\|\Delta\|^2}{2\sigma^2} + \frac{(\alpha-1)\|\Delta\|^2}{2\sigma^2} = \frac{\alpha\|\Delta\|^2}{2\sigma^2}
$$

The first equality is the definition of the Rényi divergence, in the second we input the values of $P$ and $Q$, then we open the norm, and then we calculate the expectation using the moment generating function of a Gaussian random vector $\mathbb{E}\left[ e^{\langle a, X \rangle} \right] = e^{\langle \mu, a \rangle + \frac{1}{2}\sigma^2\|a\|^2}$, and finally the log and the exponent cancel each other, and we fix things up.

## B.4. Proof of Lemma 2.6

We will start with the first part. To do it, we will show another property of the Rényi divergence. Holder's inequality states that for any $p, q \ge 1$ such that $1/p + 1/q = 1$, we get $\|fg\|_1 \le \|f\|_p \|g\|_q$. We will pick $p = \alpha, q = \frac{\alpha}{\alpha-1}, f = \frac{P}{Q^{1/q}}, g = Q^{1/q}$, and the norm to be on an arbitrary event $A$, and then:

$$
P(A) = \int_A P(x)dx \le \left( \int_A \frac{(P(x))^\alpha}{(Q(x))^{\alpha-1}} dx \right)^{\frac{1}{\alpha}} \left( \int_A Q(x)dx \right)^{\frac{\alpha-1}{\alpha}} \le \left( e^{\mathbb{D}_\alpha(P\|Q)} Q(A) \right)^{\frac{\alpha-1}{\alpha}} \le (e^\epsilon Q(A))^{\frac{\alpha-1}{\alpha}}
$$

The equality is the definition of the probability of the event, the inequality is from Holder's, next we expand the integral over everything to get the Rényi divergence, and finally we bound the Rényi divergence by $\epsilon$. We pick the event $A$ to be $A = \mathcal{O}$, and then:

$$
\mathbb{P}\left\{ \mathcal{A}(\mathcal{S}) = \mathcal{O} \right\} \le \left( e^\epsilon \mathbb{P}\left\{ \mathcal{A}(\mathcal{S}') = \mathcal{O} \right\} \right)^{1-\frac{1}{\alpha}}
$$

If $\left( e^\epsilon \mathbb{P}\left\{ \mathcal{A}(\mathcal{S}') = \mathcal{O} \right\} \right)^{1-\frac{1}{\alpha}} \le \delta$ then $\mathbb{P}\left\{ \mathcal{A}(\mathcal{S}) = \mathcal{O} \right\} \le \delta$. If $\left( e^\epsilon \mathbb{P}\left\{ \mathcal{A}(\mathcal{S}') = \mathcal{O} \right\} \right)^{1-\frac{1}{\alpha}} > \delta$ then $\left( e^\epsilon \mathbb{P}\left\{ \mathcal{A}(\mathcal{S}') = \mathcal{O} \right\} \right)^{-\frac{1}{\alpha}} < \delta^{-\frac{1}{\alpha-1}} = e^{\frac{\log\left(\frac{1}{\delta}\right)}{\alpha-1}}$, and then:

$$
\mathbb{P}\left\{ \mathcal{A}(\mathcal{S}) = \mathcal{O} \right\} \le e^{\epsilon + \frac{\log\left(\frac{1}{\delta}\right)}{\alpha-1}} \mathbb{P}\left\{ \mathcal{A}(\mathcal{S}') = \mathcal{O} \right\}
$$

If we combine both cases, we showed that:

$$
\mathbb{P}\left\{ \mathcal{A}(\mathcal{S}) = \mathcal{O} \right\} \le \max\left\{ e^{\epsilon + \frac{\log\left(\frac{1}{\delta}\right)}{\alpha-1}} \mathbb{P}\left\{ \mathcal{A}(\mathcal{S}') = \mathcal{O} \right\}, \delta \right\} \le e^{\epsilon + \frac{\log\left(\frac{1}{\delta}\right)}{\alpha-1}} \mathbb{P}\left\{ \mathcal{A}(\mathcal{S}') = \mathcal{O} \right\} + \delta
$$

Now for the second part. Since $\mathcal{A}$ is $\left(\alpha, \frac{\alpha\rho^2}{2}\right)$-RDP for every $\alpha > 1$, then it is also $\left(\frac{\alpha\rho^2}{2} + \frac{\log\left(\frac{1}{\delta}\right)}{\alpha-1}, \delta\right)$-DP, for every $\alpha > 1, \delta \in (0, 1)$. We will pick the value that minimize this expression $\alpha = 1 + \frac{1}{\rho}\sqrt{2\log\left(\frac{1}{\delta}\right)}$, and get that for every $\delta \in (0, 1)$, $\mathcal{A}$ is $\left(\frac{\rho^2}{2} + \rho\sqrt{2\log\left(\frac{1}{\delta}\right)}, \delta\right)$-DP.

# C. Proofs of Section 3

## C.1. Proof of Equation (10)

We will start with Equation (9), $\beta_t = 1 - \frac{\alpha_{t-1}}{\alpha_t}$ and $q_t = \alpha_t d_t$.

$$
\begin{aligned}
q_t &= \alpha_t d_t = \alpha_t \left(\nabla f(x_t; z_t) + (1 - \beta_t)(d_{t-1} - \nabla f(x_{t-1}; z_t))\right) = \alpha_t \left(\nabla f(x_t; z_t) + \frac{\alpha_{t-1}}{\alpha_t}(d_{t-1} - \nabla f(x_{t-1}; z_t))\right) \\
&= \alpha_t d_t + \alpha_{t-1}(d_{t-1} - \nabla f(x_{t-1}; z_t)) = q_{t-1} + \alpha_t d_t - \alpha_{t-1}\nabla f(x_{t-1}; z_t) = q_{t-1} + s_t
\end{aligned}
$$

Where at first we use the definition of $q_t$, then Equation (9), then input our choice of $\beta_t$, open the brackets and finally reuse the definition of $q_{t-1}$ and the definition of $s_t$.

# D. Proofs of Section 4

## D.1. Proof of Lemma 4.1

We will use a definitions of $\tilde{q}_t, q_t, \tilde{s}_t, s_t, \tilde{s}_{t,i}, s_{t,i}, Y_{t,i}$.

$$
\begin{aligned}
\tilde{q}_t &= \sum_{\tau=1}^{t} \tilde{s}_\tau = \sum_{\tau=1}^{t} \frac{1}{m} \sum_{i\in\mathcal{M}_\tau} \tilde{s}_{\tau,i} = \sum_{\tau=1}^{t} \frac{1}{m} \sum_{i\in\mathcal{M}_\tau} (s_{\tau,i} + Y_{\tau,i} - Y_{\tau-1,i}) \\
&= \sum_{\tau=1}^{t} \frac{1}{m} \sum_{i\in\mathcal{M}_\tau} s_{\tau,i} + \sum_{\tau=1}^{t} \frac{1}{m} \sum_{i\in\mathcal{M}_\tau} (Y_{\tau,i} - Y_{\tau-1,i}) \\
&= \sum_{\tau=1}^{t} s_\tau + \frac{1}{m} \sum_{i=1}^{M} \sum_{\tau\le t \& \tau\in\mathcal{T}_i} (Y_{\tau,i} - Y_{\tau-1,i}) = q_t + \frac{1}{m} \sum_{i=1}^{M} Y_{t,i}
\end{aligned}
$$

At first we used $\tilde{q}_t = \sum_{\tau=1}^{t} \tilde{s}_\tau$, then $\tilde{s}_\tau = \frac{1}{m}\sum_{i\in\mathcal{M}_\tau} \tilde{s}_{\tau,i}$, then $\tilde{s}_{\tau,i} = s_{\tau,i} + Y_{\tau,i} - Y_{\tau-1,i}$, then we split the sums using the linearity of the summation. For the first sum we used $s_{\tau,i} = \frac{1}{m}\sum_{i\in\mathcal{M}_\tau} s_{\tau,i}$ and then $q_t = \sum_{\tau=1}^{t} s_\tau$. For the second sum, we switched the order of summation and introduced a new notation $\mathcal{T}_i = \{t | i \in \mathcal{M}_t\}$ to help us reverse $\mathcal{M}_\tau$. Finally, while it is not very clear, the sum over $\tau$ is a telescopic sum, due to the definition of $Y_{\tau,i}$ as the last noise generated by machine $i$ at time-step $t$, and the fact that we sum over $\mathcal{T}_i$, and thus $\sum_{\tau\le t \& \tau\in\mathcal{T}_i} (Y_{\tau,i} - Y_{\tau-1,i}) = Y_{t,i}$.

## D.2. Proof of Lemma 4.2

Before we begin, we shall bound the difference between consecutive query points:

$$
x_t - x_{t-1} = x_t - \frac{\alpha_{1:t}x_t - \alpha_t w_t}{\alpha_{1:t-1}} = \frac{\alpha_t}{\alpha_{1:t-1}}(w_t - x_t)
$$

Where the first equality is due to $\alpha_{1:t}x_t = \alpha_{1:t-1}x_{t-1} + \alpha_t w_t$. Using the above enables to bound the following scaled difference:

$$
\alpha_{t-1}\|x_t - x_{t-1}\| = \left(\frac{\alpha_{t-1}\alpha_t}{\alpha_{1:t-1}}\right)\|w_t - x_t\| \le 2D
$$

Where we use that fact that $\alpha_t = t$, which implies $\alpha_{t-1}\alpha_t = 2\alpha_{1:t-1}$, as well as the bounded diameter assumption.

First part:

$$\|s_{t,i}\| = \|\alpha_t g_{t,i} - \alpha_{t-1}\tilde{g}_{t-1,i}\| \leq (\alpha_t - \alpha_{t-1})\|g_{t,i}\| + \alpha_{t-1}\|g_{t,i} - \tilde{g}_{t-1,i}\|$$
$$= (\alpha_t - \alpha_{t-1})\|\nabla f(x_t; z_{t,i})\| + \alpha_{t-1}\|\nabla f(x_t; z_{t,i}) - \nabla f(x_{t-1}; z_{t,i})\|$$
$$\leq (\alpha_t - \alpha_{t-1})G + \alpha_{t-1}L\|x_t - x_{t-1}\| \leq G + 2LD := S$$

The first equality is from the definition of $s_{t,i}$, then we use the triangle inequality, then explicitly employ the definitions of $g_{t,i}, \tilde{g}_{t-1,i}$, next we use Lipschitz and smoothness, and finally, we employ the bound on $x_t - x_{t-1}$ and $\alpha_t = t$, and the definition of $S$.

Second part: We will also define $g_t := \frac{1}{m}\sum_{i \in \mathcal{M}} g_{t,i}$ and $\tilde{g}_t := \frac{1}{m}\sum_{i \in \mathcal{M}} \tilde{g}_{t,i}$, so that $s_t = \alpha_t g_t - \alpha_{t-1}\tilde{g}_{t-1}$. The proof will follow similarly to the proof of Lemma 2.1. At first, we will fix $\mathcal{M}_t$:

$$\mathbb{E}\|s_t - \bar{s}_t\|^2 = \mathbb{E}\left\|\frac{1}{m}\sum_{i \in \mathcal{M}_t}(s_{t,i} - \bar{s}_t)\right\|^2 = \left\|\frac{1}{m}\sum_{i \in \mathcal{M}_t}(\bar{s}_{t,i} - \bar{s}_t)\right\|^2 + \mathbb{E}\left\|\frac{1}{m}\sum_{i \in \mathcal{M}_t}(s_{t,i} - \bar{s}_{t,i})\right\|^2$$

This are the same steps as the proof of Lemma 2.1. The second term is:

$$\mathbb{E}\left\|\frac{1}{m}\sum_{i \in \mathcal{M}_t}(s_{t,i} - \bar{s}_{t,i})\right\|^2 = \frac{1}{m^2}\sum_{i \in \mathcal{M}_t}\mathbb{E}\|s_{t,i} - \bar{s}_{t,i}\|^2 = \frac{1}{m^2}\sum_{i \in \mathcal{M}_t}\mathbb{E}\|\alpha_t(g_{t,i} - \bar{g}_{t,i}) - \alpha_{t-1}(\tilde{g}_{t-1,i} - \bar{g}_{t-1,i})\|^2$$
$$\leq \frac{1}{m^2}\sum_{i \in \mathcal{M}_t}\left((\alpha_t - \alpha_{t-1})\sqrt{\mathbb{E}\|g_{t,i} - \bar{g}_{t,i}\|^2} + \alpha_{t-1}\sqrt{\mathbb{E}\|(g_{t,i} - \bar{g}_{t,i}) - (\tilde{g}_{t-1,i} - \bar{g}_{t-1,i})\|^2}\right)^2$$
$$\leq \frac{1}{m^2}\sum_{i \in \mathcal{M}_t}\left((\alpha_t - \alpha_{t-1})\sigma + \alpha_{t-1}\sigma_L\|x_t - x_{t-1}\|\right)^2 \leq \frac{(\sigma + 2\sigma_L D)^2}{m} = \frac{\tilde{\sigma}^2}{m}$$

Where we used the fact the each $s_{t,i} - \bar{s}_{t,i}$ is independent with zero mean, used the definitions of $s_{t,i}, \bar{s}_{t,i}$, then we use the inequality: $\mathbb{E}\|X + Y\|^2 \leq \left(\sqrt{\mathbb{E}\|X\|^2} + \sqrt{\mathbb{E}\|Y\|^2}\right)^2$, which holds since:

$$\mathbb{E}\|X + Y\|^2 = \mathbb{E}\|X\|^2 + 2\mathbb{E}\langle X, Y\rangle + \mathbb{E}\|Y\|^2$$
$$\leq \mathbb{E}\|X\|^2 + 2\sqrt{\mathbb{E}\|X\|^2\mathbb{E}\|Y\|^2} + \mathbb{E}\|Y\|^2 = \left(\sqrt{\mathbb{E}\|X\|^2} + \sqrt{\mathbb{E}\|Y\|^2}\right)^2$$

And finally, we used Equations (4) and (5), the bounds from the first part, and the definition of $\tilde{\sigma} = \sigma + 2\sigma_L D$.

The first term also follows similar steps to the proof of Lemma 2.1:

$$\left\|\frac{1}{m}\sum_{i \in \mathcal{M}_t}(\bar{s}_{t,i} - \bar{s}_t)\right\|^2 = \frac{1}{m^2}\sum_{i=1}^{M}\sum_{j=1}^{M}\mathbb{I}\{i \in \mathcal{M}_t\}\mathbb{I}\{j \in \mathcal{M}_t\}\langle\bar{s}_{t,i} - \bar{s}_t, \bar{s}_{t,j} - \bar{s}_t\rangle$$

Taking the expectation over the randomization of $\mathcal{M}_t$ that is in the indicators, and following the same steps of the proof of

Lemma 2.1, we get:

$$
\mathbb{E}\left\|\frac{1}{m}\sum_{i\in\mathcal{M}_t}(\bar{s}_{t,i}-\bar{s}_t)\right\|^2 = \frac{M-m}{m(M-1)}\frac{1}{M}\sum_{i=1}^{M}\|\bar{s}_{t,i}-\bar{s}_t\|^2
$$

$$
=\frac{M-m}{m(M-1)}\frac{1}{M}\sum_{i=1}^{M}\|\alpha_t(\bar{g}_{t,i}-\bar{g}_t)-\alpha_t(\bar{g}_{t-1,i}-\bar{g}_{t-1})\|^2
$$

$$
\leq\frac{M-m}{m(M-1)}\left((\alpha_t-\alpha_{t-1})\sqrt{\frac{1}{M}\sum_{i=1}^{M}\|\bar{g}_{t,i}-\bar{g}_t\|^2}+\alpha_{t-1}\sqrt{\frac{1}{M}\sum_{i=1}^{M}\|(\bar{g}_{t,i}-\bar{g}_t)-(\bar{g}_{t-1,i}-\bar{g}_{t-1})\|^2}\right)^2
$$

$$
\leq\frac{M-m}{m(M-1)}\left((\alpha_t-\alpha_{t-1})\xi+\alpha_{t-1}\xi_L\|x_t-x_{t-1}\|\right)^2
$$

$$
\leq\frac{M-m}{M-1}\frac{(\xi+2\xi_L D)^2}{m}=\frac{M-m}{M-1}\frac{\tilde{\xi}^2}{m}
$$

Where at first we used the steps of the proof of Lemma 2.1, then used the definitions of $\bar{s}_{t,i},\bar{s}_t$, then used $\mathbb{E}\|X+Y\|^2\leq\left(\sqrt{\mathbb{E}\|X\|^2}+\sqrt{\mathbb{E}\|Y\|^2}\right)^2$ with empirical mean instead of expectation, and finally used Equations (6) and (7), the bounds from the first part, and the definitions of $\tilde{\xi}=\xi+2\xi_L D$.

In total, we got:

$$
\mathbb{E}\|s_t-\bar{s}_t\|^2=\frac{1}{m}\left(\tilde{\sigma}^2+\frac{M-m}{M-1}\tilde{\xi}^2\right)
$$

### D.3. Proof of Equation (15)

We will start with the definition of $\varepsilon_t$ and the update rule of $q_t$:

$$
\varepsilon_t=q_t-\alpha_t\bar{g}_t=q_{t-1}+s_t-\alpha_t\bar{g}_t=\varepsilon_{t-1}+\alpha_{t-1}\bar{g}_{t-1}+s_t-\alpha_t\bar{g}_t=\varepsilon_{t-1}+s_t-\bar{s}_t
$$

Where at first we use the definition of $\varepsilon_t$, then Equation (10), then the definition of $\varepsilon_{t-1}$, and finally the definition of $\bar{s}_t$.

### D.4. Proof of Lemma 4.3

Notice that $\varepsilon_t$ is a Martingale sequence with a difference sequence of $s_t-\bar{s}_t$. It means that:

$$
\mathbb{E}\|\varepsilon_t\|^2=\sum_{\tau=1}^{t}\mathbb{E}\|s_\tau-\bar{s}_\tau\|^2\leq\sum_{\tau=1}^{t}\frac{1}{m}\left(\tilde{\sigma}^2+\frac{M-m}{M-1}\tilde{\xi}^2\right)=\frac{t}{m}\left(\tilde{\sigma}^2+\frac{M-m}{M-1}\tilde{\xi}^2\right)
$$

Where at first we used Lemma A.2, and then used Lemma 4.2.

## E. Proofs of Section 5

### E.1. Proof of Theorem 5.1

Let us look at a single machine $i$. Let $\mathcal{S}_i,\mathcal{S}_i'$ be *neighboring datasets* of $|\mathcal{T}_i|$ samples which differ on a single data point; i.e. assume there exists $\tau^*\in\mathcal{T}_i$ such that $\mathcal{S}_i:=\{z_{t_1,i},z_{t_2,i},\ldots,z_{\tau^*,i},\ldots,z_{t_{|\mathcal{T}_i|},i}\}$ and $\mathcal{S}_i':=\{z_{t_1,i},z_{t_2,i},\ldots,z_{\tau^*,i}',\ldots,z_{t_{|\mathcal{T}_i|},i}\}$, and $z_{\tau^*}\neq z_{\tau^*}'$.
**Notation:** We will employ $s_{t,i}(\mathcal{S}_i),\tilde{s}_{t,i}(\mathcal{S}_i)$ to denote the resulting values of these quantities when we invoke our algorithm with the dataset $\mathcal{S}_i$, we will similarly denote $s_{t,i}(\mathcal{S}_i'),\tilde{s}_{t,i}(\mathcal{S}_i')$.
**Setup:** At first let's bound the sensitivity of $s_{\tau^*,i}$ meaning the difference between $s_{\tau^*,i}(\mathcal{S}_i)$ and $s_{\tau^*,i}(\mathcal{S}_i')$ conditioned on the values of $\{x_\tau\}_{\tau=1}^{\tau^*}$. The value of $x_{t+1}$ is calculated using $\tilde{s}_{t,i}$ and the other signals from the other machines, and the server sends $x_{t+1}$ to them afterwards. Thus, we may assume that prior to the computation of $s_{t,i}$, we are given $\{x_\tau\}_{\tau=1}^{t}$.

Note that given the query history $x_1, \ldots, x_{\tau^*}$, and the dataset $\mathcal{S}_i$, $s_{\tau^*,i}(\mathcal{S}_i)$ is known (respectively $s_{\tau^*,i}(\mathcal{S}_i')$ is known given the dataset $\mathcal{S}_i'$ and the query history). Also note that $s_{t,i}$ is calculated using only $x_t, x_{t-1}, z_{t,i}$, and thus for all $t \neq \tau^*$ we get $s_{t,i}(\mathcal{S}_i) = s_{t,i}(\mathcal{S}_i')$, thus we only need the sensitivity of $s_{\tau^*,i}$ (but for all possible values of $\tau^*, i$).

**Sensitivity:** Since $\|s_{\tau^*,i}\| \leq S$ (Lemma 4.2), then $\|s_{\tau^*,i}(\mathcal{S}_i) - s_{\tau^*,i}(\mathcal{S}_i')\| \leq \|s_{\tau^*,i}(\mathcal{S}_i)\| + \|s_{\tau^*,i}(\mathcal{S}_i')\| \leq 2S$.

**Bounding the Divergence:** Here we will bound the Rényi Divergence of the sequence $\{\tilde{s}_{t,i}\}_{t \in \mathcal{T}_i}$. Note that $\tilde{s}_{t,i} = s_{t,i} + y_{t,i} - Y_{t-1,i}$, and that $\forall t \in \mathcal{T}_i$, we get that $\sum_{\tau \leq t \& \tau \in \mathcal{T}_i} \tilde{s}_{\tau,i} = \sum_{\tau \leq t \& \tau \in \mathcal{T}_i} s_{\tau,i} + y_{t,i}$. It means that:

$$
\begin{aligned}
\mathbb{D}_\alpha \left( \{\tilde{s}_{t,i}(\mathcal{S}_i)\}_{t \in \mathcal{T}_i} \| \{\tilde{s}_{t,i}(\mathcal{S}_i')\}_{t \in \mathcal{T}_i} \right) =& \mathbb{D}_\alpha \left( \left\{ \sum_{\tau \leq t \& \tau \in \mathcal{T}_i} \tilde{s}_{\tau,i}(\mathcal{S}_i) \right\}_{t \in \mathcal{T}_i} \Big\| \left\{ \sum_{\tau \leq t \& \tau \in \mathcal{T}_i} \tilde{s}_{\tau,i}(\mathcal{S}_i') \right\}_{t \in \mathcal{T}_i} \right) \\
\leq& \sum_{t \in \mathcal{T}_i} \mathbb{D}_\alpha \left( \sum_{\tau \leq t \& \tau \in \mathcal{T}_i} \tilde{s}_{\tau,i}(\mathcal{S}_i) \Big\| \sum_{\tau \leq t \& \tau \in \mathcal{T}_i} \tilde{s}_{\tau,i}(\mathcal{S}_i') \right) \\
=& \sum_{t \in \mathcal{T}_i} \frac{\alpha \left\| \sum_{\tau \leq t \& \tau \in \mathcal{T}_i} s_{\tau,i}(\mathcal{S}_i) - \sum_{\tau \leq t \& \tau \in \mathcal{T}_i} s_{\tau,i}(\mathcal{S}_i') \right\|^2}{2\sigma_t^2} \\
=& \alpha \sum_{t \in \mathcal{T}_i \& t \geq \tau^*} \frac{\|s_{\tau^*,i}(\mathcal{S}_i) - s_{\tau^*,i}(\mathcal{S}_i')\|^2}{2\sigma_t^2} \leq 2\alpha S^2 \sum_{t \in \mathcal{T}_i} \frac{1}{\sigma_t^2}
\end{aligned}
$$

Where at first we use the post-processing property (Lemma A.4), then to composition rule (Lemma A.3), then we use the result of the Gaussian mechanism (Lemma 2.3), then use the fact that $s_{\tau,i}(\mathcal{S}_i) = s_{\tau,i}(\mathcal{S}_i')$ for all $\tau \neq \tau^*$, and finally we bound each difference by $2S$ and sum more terms. We got that our algorithm is $2S^2 \sum_{t \in \mathcal{T}_i} \frac{1}{\sigma_t^2}$-zCDP, meaning it is $\frac{\rho_i^2}{2}$-zCDP if $\rho_i = 2S\sqrt{\sum_{t \in \mathcal{T}_i} \frac{1}{\sigma_t^2}}$.

### E.2. Proof of Theorem 5.2

Start by bounding the excess loss $\mathcal{R}_t$:

$$
\begin{aligned}
\alpha_{1:t} \mathcal{R}_t =& \alpha_{1:t} \left( \mathbb{E}[f(x_t)] - f(x^*) \right) \leq \sum_{\tau=1}^{t} \mathbb{E}[\alpha_\tau \langle \nabla f(x_\tau), w_\tau - x^* \rangle] = \sum_{\tau=1}^{t} \mathbb{E} \langle \tilde{q}_\tau - \varepsilon_\tau - Y_\tau, w_\tau - x^* \rangle \\
=& \sum_{\tau=1}^{t} \mathbb{E} \langle \tilde{q}_\tau, w_\tau - x^* \rangle + \sum_{\tau=1}^{t} \mathbb{E} \langle \varepsilon_\tau + Y_\tau, x^* - w_\tau \rangle \\
=& \sum_{\tau=1}^{t} \mathbb{E} \langle \tilde{q}_\tau, w_{\tau+1} - x^* \rangle + \sum_{\tau=1}^{t} \mathbb{E} \langle \tilde{q}_\tau, w_\tau - w_{\tau+1} \rangle + \sum_{\tau=1}^{t} \mathbb{E} \langle \varepsilon_\tau + Y_\tau, x^* - w_\tau \rangle \\
\leq& \frac{D^2}{2\eta} - \frac{1}{2\eta} \sum_{\tau=1}^{t} \mathbb{E} \|w_\tau - w_{\tau+1}\|^2 + \sum_{\tau=1}^{t} \mathbb{E} \langle \tilde{q}_\tau, w_\tau - w_{\tau+1} \rangle + \sum_{\tau=1}^{t} \mathbb{E} \langle \varepsilon_\tau + Y_\tau, x^* - w_\tau \rangle \\
\leq& \frac{D^2}{2\eta} + \sum_{\tau=1}^{t} \alpha_\tau \mathbb{E} \langle \nabla f(x_\tau) - \nabla f(x^*), w_\tau - w_{\tau+1} \rangle - \frac{1}{2\eta} \|w_\tau - w_{\tau+1}\|^2 \\
& + \underbrace{\sum_{\tau=1}^{t} \mathbb{E} \langle \alpha_\tau \nabla f(x^*), w_\tau - w_{\tau+1} \rangle}_{(A)} + \underbrace{\sum_{\tau=1}^{t} \mathbb{E} \langle \varepsilon_\tau, x^* - w_{\tau+1} \rangle}_{(B)} + \underbrace{\sum_{\tau=1}^{t} \mathbb{E} \langle Y_\tau, x^* - w_{\tau+1} \rangle}_{(C)}
\end{aligned}
$$

The first inequality is the anytime theorem (Theorem A.1), then we use $\tilde{q}_t = q_t + \varepsilon_t + Y_t$, split it into two sums, split the first sum using $w_{t+1}$, bound the first sum using Lemma A.5, and finally we add and subtract $\nabla f(x^*)$ and reuse the definition of $\tilde{q}_t$ and add it to the other sums. We will now bound the bottom terms:

**Bounding (A)**: This term can be written as follows:

$$(\text{A}) := \sum_{\tau=1}^{t} \mathbb{E}\langle \alpha_\tau \nabla f(x^*), w_\tau - w_{\tau+1}\rangle = \sum_{\tau=1}^{t}(\alpha_\tau - \alpha_{\tau-1})\mathbb{E}\langle \nabla f(x^*), w_\tau\rangle - \alpha_t \mathbb{E}\langle \nabla f(x^*), w_{t+1}\rangle$$

$$= \sum_{\tau=1}^{t}(\alpha_\tau - \alpha_{\tau-1})\mathbb{E}\langle \nabla f(x^*), w_\tau - w_{t+1}\rangle \leq \sum_{\tau=1}^{t}(\alpha_\tau - \alpha_{\tau-1})\|\nabla f(x^*)\|\,\mathbb{E}[\|w_\tau - w_{t+1}\|]$$

$$\leq D\|\nabla f(x^*)\|\sum_{\tau=1}^{t}(\alpha_\tau - \alpha_{\tau-1}) = \alpha_t D\|\nabla f(x^*)\| = \alpha_t D G^*$$

The first equality is rearrangement of the sum while defining $\alpha_0 = 0$, then we put the last term into the sum, and use Cauchy-Schwartz. Finally, we use the diameter bound and telescope the sum. Note that we use the definition $G^* := \|\nabla f(x^*)\|$, and that $G^* \in [0, G]$.

**Bounding (B)**: This term is bounded as follows:

$$(\text{B}) := \sum_{\tau=1}^{t} \mathbb{E}\langle \varepsilon_\tau, x^* - w_{\tau+1}\rangle \leq \sum_{\tau=1}^{t}\mathbb{E}[\|\varepsilon_\tau\| \cdot \|x^* - w_{\tau+1}\|]$$

$$\leq D\sum_{\tau=1}^{t}\sqrt{\mathbb{E}\|\varepsilon_\tau\|^2} \leq D\sum_{\tau=1}^{t}\sqrt{\frac{\left(\tilde{\sigma}^2 + \tilde{\xi}^2\right)\tau}{m}} \leq \frac{D\sqrt{\tilde{\sigma}^2 + \tilde{\xi}^2}\,t^{1.5}}{\sqrt{m}}$$

The first inequality is Cauchy-Schwartz, and the second is using the bounded diameter and Jensen's inequality w.r.t. concave function $\sqrt{x}$, then we use Lemma 4.3, and finally increase $\tau$ to $t$.

**Bounding (C)**: This term is bounded as follows:

$$(\text{C}) := \sum_{\tau=1}^{t}\mathbb{E}\langle Y_\tau, x^* - w_{\tau+1}\rangle = \sum_{\tau=1}^{t}\frac{1}{m}\sum_{i=1}^{M}\mathbb{E}\langle Y_{\tau,i}, x^* - w_{\tau+1}\rangle$$

$$= \frac{1}{m}\sum_{i=1}^{M}\sum_{\tau=1}^{t}\sum_{s=1}^{\tau}p(1-p)^{\tau-s}\mathbb{E}\langle y_{s,i}, x^* - w_{\tau+1}\rangle = \frac{p}{m}\sum_{i=1}^{M}\sum_{\tau=1}^{t}\sum_{s=1}^{\tau}(1-p)^{\tau-s}\mathbb{E}\langle y_{s,i}, w_s - w_{\tau+1}\rangle$$

$$= \frac{p}{m}\sum_{i=1}^{M}\sum_{\tau=1}^{t}\sum_{s=1}^{\tau}(1-p)^{\tau-s}\sum_{r=s}^{\tau}\mathbb{E}\langle y_{s,i}, w_r - w_{r+1}\rangle$$

$$= \frac{p}{m}\sum_{i=1}^{M}\sum_{r=1}^{t}\sum_{s=1}^{r}(1-p)^{r-s}\mathbb{E}\langle y_{s,i}, w_r - w_{r+1}\rangle\sum_{\tau=r}^{t}(1-p)^{\tau-r}$$

$$\leq \frac{1}{m}\sum_{i=1}^{M}\sum_{r=1}^{t}\sum_{s=1}^{r}(1-p)^{r-s}\mathbb{E}\langle y_{s,i}, w_r - w_{r+1}\rangle = \sum_{r=1}^{t}\mathbb{E}\left\langle\left(\frac{1}{m}\sum_{i=1}^{M}\sum_{s=1}^{r}(1-p)^{r-s}y_{s,i}\right), w_r - w_{r+1}\right\rangle$$

$$\leq \frac{1}{4\eta}\sum_{r=1}^{t}\mathbb{E}\|w_r - w_{r+1}\|^2 + \eta\sum_{r=1}^{t}\mathbb{E}\left\|\frac{1}{m}\sum_{i=1}^{M}\sum_{s=1}^{r}(1-p)^{r-s}y_{s,i}\right\|^2$$

At first we write $Y_\tau$ as shown in Lemma 4.1, then we know that $Y_{\tau,i}$ is actually $y_{s,i}$, where $s$ is a geometric random variable that starts at $\tau$ downwards with probability $p = \frac{m}{M}$. $y_{s,i}$ is independent of everything up to the time step $s$, so we replace $x^*$ with $w_s$, divide $w_s - w_{\tau+1}$ into a sum of differences, then rearrange the order of summation, and the sum in $\tau$ is a geometric sum bounded by $\frac{1}{p}$. We put the sum of $r$ outside and use Cauchy-Schwartz with $\frac{1}{4\eta}$ and $\eta$ to split it into two sums. We will

now bound the second sum:

$$\eta \sum_{r=1}^{t} \mathbb{E} \left\| \frac{1}{m} \sum_{i=1}^{M} \sum_{s=1}^{r} (1-p)^{r-s} y_{s,i} \right\|^2 \leq \frac{\eta}{m^2} \sum_{r=1}^{t} \sum_{i=1}^{M} \sum_{s=1}^{r} (1-p)^{2(r-s)} \mathbb{E} \|y_{s,i}\|^2$$

$$= \frac{\eta}{m^2} \sum_{i=1}^{M} \sum_{s=1}^{t} \mathbb{E} \|y_{s,i}\|^2 \sum_{r=s}^{t} (1-p)^{2(r-s)} \leq \frac{\eta}{m^2 p(2-p)} \sum_{i=1}^{M} \sum_{s=1}^{t} \mathbb{E} \|y_{s,i}\|^2$$

$$= \frac{\eta}{m^2 p(2-p)} \sum_{i=1}^{M} \sum_{s=1}^{t} \frac{4S^2 d\,(1+\log T)}{\rho^2} \mathbb{E}\left[N_{s,i}\right] = \frac{4S^2 \eta d\,(1+\log T)}{\rho^2 m^2 p(2-p)} \sum_{i=1}^{M} \sum_{s=1}^{t} (1+p(s-1))$$

$$= \frac{4S^2 \eta M d\,(1+\log T)}{\rho^2 m^2 p(2-p)} \left(t + p\frac{t(t-1)}{2}\right) = \frac{2S^2 \eta d\,(1+\log T)}{\rho^2 m p^2 (2-p)} \left(pt^2 + (2-p)t\right) \leq \frac{2S^2 \eta t d\,(1+\log T)}{\rho^2 m p^2} (pt+1)$$

Each $y_{s,i}$ is independent of each other and with zero mean, so the variance of the sum is the sum of variances, with the factors squared. We rearrange the summation order and the sum of $r$ is a geometric sum that is bounded by $\frac{1}{p(2-p)}$, and then we use the variance of $y_{s,i}$, which is proportional to the number of time steps of machine $i$ played up to the time step $s$, which we call $N_{s,i}$. $N_{s,i} - 1$ is binomial random variables with $s-1$ trials with probability $p$, so we input the expectation of it, then we solve both remaining sums. We input $M = \frac{m}{p}$, and finally we cancel the two $2-p$ and bound it from below by 1.

In total, we get:

$$\alpha_{1:t} \mathcal{R}_t \leq \sum_{\tau=1}^{t} \alpha_\tau \mathbb{E} \langle \nabla f(x_\tau) - \nabla f(x^*), w_\tau - w_{\tau+1} \rangle - \frac{1}{4\eta} \|w_\tau - w_{\tau+1}\|^2$$

$$+ \frac{D^2}{2\eta} + \alpha_t D G^* + \frac{D\sqrt{\tilde{\sigma}^2 + \tilde{\xi}^2} t^{1.5}}{\sqrt{m}} + \frac{2S^2 \eta t d\,(1+\log T)}{\rho^2 m p^2} (pt+1)$$

$$\leq \eta \sum_{\tau=1}^{t} \alpha_\tau^2 \mathbb{E} \|\nabla f(x_\tau) - \nabla f(x^*)\|^2 + \frac{D^2}{2\eta} + \alpha_t D G^* + \frac{D\sqrt{\tilde{\sigma}^2 + \tilde{\xi}^2} t^{1.5}}{\sqrt{m}} + \frac{2S^2 \eta t d\,(1+\log T)}{\rho^2 m p^2} (pt+1)$$

$$\leq 4\eta L \sum_{\tau=1}^{t} \alpha_{1:\tau} \mathcal{R}_\tau + \frac{D^2}{2\eta} + \alpha_t D G^* + \frac{D\sqrt{\tilde{\sigma}^2 + \tilde{\xi}^2} t^{1.5}}{\sqrt{m}} + \frac{2S^2 \eta t d\,(1+\log T)}{\rho^2 m p} \left(t + \frac{1}{p}\right)$$

Where at first we inputted what we got, used Cauchy-Schwartz, and finally bounded $\alpha_\tau^2 \leq 2\alpha_{1:\tau}$ and $\mathbb{E}\|\nabla f(x_\tau) - \nabla f(x^*)\|^2 \leq 2L\mathcal{R}_\tau$ (Lemma A.6). If we enforce $\eta \leq \frac{1}{8LT}$ we can use Lemma A.7 to get:

$$\alpha_{1:T} \mathcal{R}_T \leq \frac{D^2}{\eta} + 2\alpha_T D G^* + \frac{2D\sqrt{\tilde{\sigma}^2 + \tilde{\xi}^2} T^{1.5}}{\sqrt{m}} + \frac{4S^2 \eta T d\,(1+\log T)}{\rho^2 m p} \left(T + \frac{1}{p}\right)$$

Then after dividing by $\alpha_{1:T}$ and using $\frac{1}{p} = \frac{M}{m} \leq T$, we get:

$$\mathcal{R}_T \leq \frac{2D^2}{\eta T^2} + \frac{4D G^*}{T} + \frac{4D\sqrt{\tilde{\sigma}^2 + \tilde{\xi}^2}}{\sqrt{mT}} + \frac{16S^2 \eta M d\,(1+\log T)}{\rho^2 m^2}$$

Picking $\eta = \min\left\{ \frac{\rho D m}{2ST\sqrt{2Md(1+\log T)}}, \frac{1}{8LT} \right\}$, we get:

$$\mathcal{R}_T \leq \frac{4D(G^* + 4LD)}{T} + \frac{4D\sqrt{\tilde{\sigma}^2 + \tilde{\xi}^2}}{\sqrt{mT}} + \frac{8DS\sqrt{2Md\,(1+\log T)}}{\rho m T}$$

## F. Trusted Server

The algorithm for the trusted server case is almost the same as the one in (Reshef & Levy, 2024). The only difference is that we use $m$ machines instead of $M$ at each time-step, and that we let the server hold $q_t$, instead of each machine holding its own $q_{t,i}$. Algorithm 2 depicts our approach.

---

**Algorithm 2** DP-$\mu^2$-FL for Trusted Server

---

**Inputs:** #iterations $T$, #machines in total $M$, #machines per step $m$, initial point $x_0$, learning rate $\eta > 0$, importance weights $\{\alpha_t > 0\}$, noise distributions $\{P_t = \mathcal{N}(0, I\sigma_t^2)\}$, per-machine $i \in [M]$ a dataset of samples $\mathcal{S}_i = \{z_{1,i}, \ldots, z_{T,i}\}$

**Initialize:** set $w_1 = x_1 = x_0$, and $q_0 = 0$

**for** $t = 1, \ldots, T$ **do**
    Choose $\mathcal{M}_t \subseteq M$ with $|\mathcal{M}_t| = m$
    **for** every Machine $i \in \mathcal{M}_t$ **do**
        **Actions of Machine $i$:**
        Retrieve $z_{t,i}$ from $\mathcal{S}_i$, compute $g_{t,i} = \nabla f(x_t; z_{t,i})$, and $\tilde{g}_{t-1,i} = \nabla f(x_{t-1}; z_{t,i})$
        Update $s_{t,i} = \alpha_t g_{t,i} - \alpha_{t-1}\tilde{g}_{t-1,i}$
    **end for**
    **Actions of Server:**
    Aggregate $s_t = \frac{1}{m}\sum_{i \in \mathcal{M}_t} s_{t,i}$
    Update $q_t = q_{t-1} + s_t$
    Draw $Y_t \sim \mathcal{N}(0, I\sigma_t^2)$
    Update $\tilde{q}_t = q_t + Y_t$
    Update $w_{t+1} = \Pi_{\mathcal{K}}(w_t - \eta\tilde{q}_t)$
    Update $x_{t+1} = \left(1 - \frac{\alpha_{t+1}}{\alpha_{1:t+1}}\right)x_t + \frac{\alpha_{t+1}}{\alpha_{1:t+1}}w_{t+1}$
**end for**
**Output:** $x_T$

---

### F.1. Privacy Guarantees

Here we establish the privacy guarantees of Algorithm 1. Concretely, the following theorem shows how does the privacy of our algorithm depends on the variances of injected noise $\{y_{t,i}\}_t$.

**Theorem F.1.** *Let $\mathcal{K} \subset \mathbb{R}^d$ be a convex set of diameter $D$, and $\{f_i(\cdot; z)\}_{z \in \mathcal{Z}_i}$ be a family of convex G-Lipschitz and L-smooth functions, and $SD = G + 2LD$. Then invoking Algorithm 2 with number of participating machines $|\mathcal{M}_t| = m$, noise distributions $Y_t \sim P_{t,i} = \mathcal{N}(0, I\sigma_t^2)$, and any learning rate $\eta > 0$, the resulting sequences $\{x_t\}_t$ is $\frac{\rho^2}{2}$-zCDP, where:*
$$\rho = \frac{2S}{m}\sqrt{\sum_{t=1}^{T} \frac{1}{\sigma_t^2}}.$$

The proof is the same as the one in (Reshef & Levy, 2024).

*Proof Sketch.* First, assume that $\mathcal{S}_i$ and $\mathcal{S}_i'$ are neighboring datasets, meaning that there exists only a single time-step $\tau^* \in \mathcal{T}_i$ where they differ, i.e. that $z_{\tau^*,i} \neq z'_{\tau^*,i}$.

Then, we use the post-processing property of privacy (Lemma A.4) to say that the privacy of $\{x_t\}_t$ is bounded by the privacy of $\{\tilde{q}_t\}_t$. Using the composition rule (Lemma A.3), we bound the privacy of them with the sum of the privacy of each individual member. We may use Lemma 2.3 and obtain that $\tilde{q}_t$ is $\frac{\Delta_t^2}{2\sigma_t^2}$-zCDP. Using the bound $\|s_{\tau,i}\| \leq S$, from Lemma 4.2, we show that $\Delta_t \leq \frac{2S}{m}$. Thus, we are $\frac{2S^2}{m^2\sigma_t^2}$-zCDP. Using the above together, we get that $\{x_t\}_t$ is $\frac{2S^2}{m^2}\sum_{t=1}^{T}\frac{1}{\sigma_t^2}$-zCDP. $\square$

### F.2. Convergence Guarantees

Here we establish the convergence guarantees of Algorithm 2.

**Theorem F.2.** *Let $\mathcal{K} \subset \mathbb{R}^d$ be a convex set of diameter $D$ and $\{f_i(\cdot; z)\}_{i \in [M], z \in \mathcal{Z}_i}$ be a family of G-Lipschitz and L-smooth functions over $\mathcal{K}$, with $\sigma, \xi \in [0, G], \sigma_L, \xi_L \in [0, L]$, and let $\mathcal{M}_t$ be a subset of $[M]$ of size $m$, define $G^* := \nabla f(x^*)$, where $x^* = \arg\min_{x \in \mathcal{K}} f(x)$, and $S := G + 2LD, \tilde{\sigma} := \sigma + 2\sigma_L D, \tilde{\xi} := \xi + 2\xi_L D$, moreover let $T \in \mathbb{N}, \rho > 0$.*

*Then, upon invoking Algorithm 2 with $\alpha_t = t$, $\eta = \min\left\{\frac{\rho Dm}{2ST\sqrt{d}}, \frac{1}{4LT}\right\}$, and $\sigma_t^2 = \frac{4S^2T}{\rho^2 m^2}$, with $N_{t,i}$ being the number of time steps that machine $i$ participated up to time step $t$, and any starting point $x_1 \in \mathcal{K}$ and datasets $\{\mathcal{S}_i \in \mathcal{Z}_i^T\}_{i \in [M]}$, then Algorithm 2 satisfies $\frac{\rho^2}{2}$-zCDP w.r.t the query point, i.e. $\{x_t\}_t$.*

*Furthermore, if $\mathcal{S}_i$ consists of i.i.d. samples from a distribution $\mathcal{D}_i$ for all $i \in [M]$, and $\mathcal{M}_i$ are also chosen uniformly and i.i.d then Algorithm 2 guarantees:*

$$\mathcal{R}_T := \mathbb{E}\left[f(x_T)\right] - \min_{x \in \mathcal{K}} f(x) \leq 4D \left( \frac{G^* + 2LD}{T} + \frac{\sqrt{\tilde{\sigma}^2 + \tilde{\xi}^2}}{\sqrt{mT}} + \frac{2S\sqrt{d}}{\rho mT} \right)$$

The proof is the same as the one in (Reshef & Levy, 2024). Notably, the above bounds are optimal.

*Proof Sketch.* The privacy guarantees follow directly from Theorem 5.1, and our choice of $\sigma_t^2$:

$$2S^2 \sum_{t=1}^{T} \frac{1}{\sigma_t^2} = \frac{2S^2}{m^2} \sum_{t=1}^{T} \frac{\rho^2 m^2}{4S^2 T} = \frac{\rho^2}{2T} \sum_{t=1}^{T} 1 = \frac{\rho^2}{2}$$

Regrading convergence, in the spirit of $\mu^2$-SGD analysis (Levy, 2023; Reshef & Levy, 2024), we bound the excess loss using the anytime theorem (Theorem A.1), rewrite the expression to get to the form of Lemma A.5 and use it to bound, and separate the terms of $\varepsilon_t$ and $Y_t$. We already bounded $\varepsilon_t$ in Lemma 4.3, though we bound $\frac{M-m}{M-1} \leq 1$, and now the bound on $Y_t$ is easy, since the sequence $\{Y_t\}_t$ is i.i.d., so we can use the technique of the proof in (Reshef & Levy, 2024). We then input all the bounds, bound the gradient using the excess loss with Lemma A.6, to get a bound of the excess loss using the previous excess losses. Using Lemma A.7 we get the final bound on the excess loss. By inputting our chosen $\eta$, that minimize this expression, we get our bound. $\qquad\square$

## G. Analysis of Alternative Method

Here we provide a simplified analysis which demonstrates the suboptimality of the approach we present in Equation (12). We show that the delayed updated lead to an excessive error of the gradient estimates which translates to a degraded bound, even in the non private case.

We start by bounding the error of gradient estimate, $\varepsilon_t$, similarly to Lemma 4.3. At first, we will define $\varepsilon_{t,i} := q_{t,i} - \alpha_t \nabla f_i(x_t)$, and notice that:

$$\varepsilon_t = \frac{1}{m} \sum_{i \in \mathcal{M}_t} \varepsilon_{t,i} + \alpha_t \left( \frac{1}{m} \sum_{i \in \mathcal{M}_t} \nabla f_i(x_t) - \nabla f(x_t) \right)$$

If we ignore heterogeneity for simplicity, we get:

$$\mathbb{E}\left\|\varepsilon_t\right\|^2 = \frac{1}{m^2} \sum_{i \in \mathcal{M}_t} \mathbb{E}\left\|\varepsilon_{t,i}\right\|^2$$

Now let us look at $\varepsilon_{t,i}$. Similarly to Equation (15), we can write the update step of $\varepsilon_{t,i}$ as:

$$\varepsilon_{t,i} = \varepsilon_{t-\tau,i} + \alpha_t(\nabla f_i(x_t; z_t) - \nabla f_i(x_t)) - \alpha_{t-\tau}(\nabla f_i(x_{t-\tau}; z_t) - \nabla f_i(x_{t-\tau}))$$

We can see that $\varepsilon_{t,i}$ is a martingale given the time-steps machine $i$ participates $\mathcal{T}_i$. Similarly to Lemma 4.2:

$$\mathbb{E}\left\|\alpha_t(\nabla f_i(x_t; z_t) - \nabla f_i(x_t)) - \alpha_{t-\tau}(\nabla f_i(x_{t-\tau}; z_t) - \nabla f_i(x_{t-\tau}))\right\|^2$$

$$\leq ((\alpha_t - \alpha_{t-\tau})\sigma + \alpha_{t-\tau}\sigma_L \left\|x_t - x_{t-\tau}\right\|)^2 \leq \tilde{\sigma}^2 \tau^2$$

Since we know that:

$$\alpha_{t-\tau}\left\|x_t - x_{t-\tau}\right\| \leq \alpha_{t-\tau}\frac{\alpha_{t-\tau+1:t}}{\alpha_{1:t}}D = \frac{(t-\tau)(2t-\tau+1)\tau}{(t+1)t}D \leq 2\tau D$$

Note that $\tau$ is a geometric random variable with probability $p = \frac{m}{M}$. Nevertheless, to simplify the analysis let us look at the "average case", where $\tau = \frac{M}{m}$:

$$\mathbb{E}\left\|\varepsilon_{t,i}\right\|^2 \leq \mathbb{E}\left\|\varepsilon_{t-\tau,i}\right\|^2 + \tilde{\sigma}^2 \tau^2 \leq \tilde{\sigma}^2 \sum_{k=1}^{t/\tau} \tau^2 = \tilde{\sigma}^2 t\tau = \frac{Mt}{m}\tilde{\sigma}^2$$

Thus, in total, we get:

$$\mathbb{E}\left\|\varepsilon_t\right\|^2 = \frac{1}{m^2}\sum_{i\in\mathcal{M}_t}\mathbb{E}\left\|\varepsilon_{t,i}\right\|^2 \le \frac{1}{m^2}\sum_{i\in\mathcal{M}_t}\frac{Mt}{m}\tilde{\sigma}^2 = \frac{Mt}{m^2}\tilde{\sigma}^2$$

It is $\frac{M}{m}$ times larger than what we get in Lemma 4.3. Plugging these bounds back into the bound for $\mathcal{R}_t$, in a similar manner to what we do in the proof of Theorem 5.2 (see Appendix E.2) leads to the following additive term in the error bound,

$$\frac{2}{\alpha_{1:T}}D\sum_{\tau=1}^{T}\sqrt{\mathbb{E}\left\|\varepsilon_\tau\right\|^2} \le \frac{4D}{T^2}\sqrt{\frac{M}{m}}\sum_{\tau=1}^{T}\sqrt{\frac{\left(\tilde{\sigma}^2+\tilde{\xi}^2\right)\tau}{m}} \le \frac{4D\sqrt{\tilde{\sigma}^2+\tilde{\xi}^2}}{\sqrt{mT}}\sqrt{\frac{M}{m}}$$

This substantiates the degradation.

