# OpenReview forum: "Privacy-Preserving Federated Convex Optimization: Balancing Partial-Participation and Efficiency via Noise Cancellation"
_ICML.cc/2025/Conference — ICML 2025 poster_

### Official Review · Reviewer_Aidn · 2025-03-07

**Overall Recommendation:** 3

**Summary:**

The paper proposes DP federated learning algorithms that allows partial participation based on the DP $\mu^2$ SGD algorithms from (Reshef & Levy, 2024). The objective is to minimize the average of the population risks across all clients. The authors consider both cases of trusted and untrusted servers and achieve optimal convergence rate and linear computation efficiency. The case of trusted server is just a straightforward extension to (Reshef & Levy, 2024) where the queries are updated on the server side. Meanwhile, in the case of trusted server, the authors propose a technique of noise cancellation such that the injected noise at the server will be i.i.d and enjoy some concentration properties.

**Claims And Evidence:**

All the claims make sense to me and are supported by convincing proofs.

**Essential References Not Discussed:**

Please see my comments in section "Relation To Broader Scientific Literature"

**Experimental Designs Or Analyses:**

See my reviews in the section "Methods And Evaluation Criteria".

**Methods And Evaluation Criteria:**

There are no empirical demonstrations in the paper. It will be greatly appreciated if the authors can include some experiments and comparisons to other DP FL methods.

**Other Comments Or Suggestions:**

N/A

**Other Strengths And Weaknesses:**

N/A

**Questions For Authors:**

N/A

**Relation To Broader Scientific Literature:**

The contributions of this paper, especially compared with prior works, are not clear to me. The contribution of the trusted server case is trivial given the extension from (Reshef & Levy, 2024)  is straightforward.

For the case of non-trusted server, the authors mention in the paper that (Lowy & Razaviyayn, 2023) achieves the same rate but with $|S|^{3/2}$ computations. However, given the problem of LDP SCO is a well studied area, I am wondering if there are any other related works (even in centralized setting) that achieves optimal rate and linear computational complexity. If so, how is this work compared with them?

**Theoretical Claims:**

I took a quick look at the proofs but didn't go through them thoroughly. The proofs look reasonable and seem correct to me.

---

> ### Author Rebuttal · Authors · 2025-04-01
>
> Thank you for your positive review and for your comments.
>
> We address all of your concerns and kindly ask you to update your score accordingly.
>
> ---
> **Q: Adding experiments**
> **A:** We have now added experiments that demonstrate the applicability of our approach and corroborate our theoretical guarantees (Logistic regression over MNIST). Please see our response to Reviewer **kvrv**.
>
> ---
>
> **Q: Novelty over the full-participation paper [Reshef and Levy, 2024]**
> **A:** The **trusted server setting** is indeed quite simple — as explicitly mentioned in our paper.
> But this is **not the case** for the **untrusted-server setting**, which is our **main contribution**.
>
> We understand that our approach seems simple in hindsight. However:
>
> - In **Section 4.1** of our paper, we depict **two natural extensions** of [Reshef and Levy, 2024] that completely **fail to achieve optimal guarantees**.
> - As shown in our analysis in **Appendices D and E**, the combination of **partial participation** and **correlated noise**:
>      - substantially complicates the analysis
>      - introduces challenges well beyond those in [Reshef and Levy, 2024]
>
> -Moreover, our approach is:
>     (i) the **first to achieve optimal generalization guarantees** in the partial-participation setting
>     (ii) the **first to achieve linear computational complexity** in this setting
>
> -Finally, we believe that:
>     (i) a **simple algorithm is an advantage**, not a disadvantage
>     (ii) while our approach is simple in hindsight, it was **very challenging to develop and analyze this non-trivial idea**
>
> ---
>
> **Q: Other related works**
> **A:** Let us re-emphasize the relation to previous works:
>
> - For the case **where machines trust each other**:
>      - The work of [Feldman et al. 2020] achieves **optimal guarantees** with **linear computational complexity**, but **only for the last iterate**.
>   - Their technique **does not extend** to the case where machines do not trust each other, since they **only maintain privacy of the last iterate**, while all intermediate iterates are **non-private**.
>      - Conversely, our approach works when machines **do not trust each other** (regardless of whether they trust the server), and still obtains optimal guarantees and complexity.
>
> - For the case **where machines do not trust each other**, beyond the work of [Lowy & Razaviyayn, 2023] that requires  $O(|S|^{3/2})$ complexity, there is a recent work of [Gao et al. 2024] brought to our attention by Reviewer *kvrv*. However:
>
>    - In the **full-participation** case, [Gao et al. 2024] achieves **optimal generalization** with a complexity of $O(|S|^{9/8})$, which improves over [Lowy & Razaviyayn, 2023], but is **still suboptimal** compared to our work.
>
>    - In the **partial-participation** case, [Gao et al. 2024] achieves **suboptimal guarantees** (see below), and still requires **superlinear computation** in $|S|$.
>
>    -Conversely, **we obtain optimal performance for the partial-participation case**, and enjoy **linear computational complexity** in $|S|$.
>
> **Elaborate Comparison to [Gao et al. 2024]**
>
> The work of [Gao et al. 2024] indeed provides guarantees for DP-FL learning.
>
> **Regarding guarantees:**
>
> - In the **full-participation** setting:
>     - [Gao et al. 2024] achieves **optimal DP generalization** (i.e., population loss guarantees)
>    - However, their approach requires **super-linear computational complexity**, i.e., $O(|S|^{9/8})$, where $|S|$ is the total number of  samples used (see Eq. (7) in their paper)
>    - Note that in their notation:
>     $|S| = nN$, where:
>        - $N$ = total number of machines
>        - $n$ = number of samples per machine
>
> - **Comparably, in this setting** we:
>      - achieve **optimal DP generalization**
>      - require **only linear computational complexity** in $|S|$
>      - In our notation, $|S| = n$ denotes the **total number of datapoints used by all machines** during training
>
> - In the **partial-participation** setting:
>      - [Gao et al. 2024] still uses $|S| = nN$ (even when only $M \leq N$ machines are used per round; see Alg. 1 and Alg. 2)
>      - Theorem C.1 (Eq. (17)) in their paper shows a convergence rate of:
>     $$
>     O\left(\frac{1}{\sqrt{Mn}} + \frac{\sqrt{d}}{\epsilon \sqrt{M}n}\right)
>     $$
>     -Translating to total data $|S| = nN$, this becomes:
>     $$
>     O\left(\frac{N/M}{\sqrt{|S|}} + \frac{\sqrt{d}N/\sqrt{M}}{\epsilon |S|}\right)
>     $$
>     which is **suboptimal**
>   - Moreover, their approach still requires **superlinear computation** in $|S|$
>
> - **Conversely, in our partial-participation setting**, we:
>   - obtain **optimal guarantees** of:
>     $$
>     O\left(\frac{1}{\sqrt{|S|}} + \frac{\sqrt{d}\sqrt{M}}{|S|}\right)
>     $$
>   - achieve **linear computational complexity** in $|S|$
>   - Note: in our notation, $M$ = total number of machines
>
> We shall add this comparison to the paper.

---

### Official Review · Reviewer_T3sK · 2025-03-13

**Overall Recommendation:** 1

**Summary:**

The paper addresses the challenge of applying DP in FL when only a subset of clients are active per iteration.

It mainly improves on a previous paper called 'Private and Federated Stochastic Convex Optimization: Efficient Strategies for Centralized Systems', in which all settings are same but the clients are all active.

The authors argue that the previous proposed full participation approach does not extent well to partial participation case.  The main challenge is that the partial participation case will accumulate noise in the global estimate.

To overcome this, the authors propose a novel noise-cancellation mechanism. Specifically, they introduce a method where the added noise at each round is adjusted by subtracting the noise from the previous round (i.e., the noise is defined as Y^(t+1)-Y^(t) ) This approach prevents cumulative noise growth, which can otherwise degrade model accuracy and convergence.

**Claims And Evidence:**

The claim is supported by rigorous theoretical proofs

**Essential References Not Discussed:**

Noise cancellation in DP related literature are not discussed. E.g.,  'DPNCT: A Differential Private Noise Cancellation Scheme for Load Monitoring and Billing for Smart Meters'

**Experimental Designs Or Analyses:**

No, there is no experimental validation

**Methods And Evaluation Criteria:**

While it make sense to address the challenge of cumulative noise by introducing noise cancellation mechanism, it is not a novel technique proposed firstly in federated convex optimization, but has been applied in other applications such as 'DPNCT: A Differential Private Noise Cancellation Scheme for Load Monitoring and Billing for Smart Meters'

**Other Comments Or Suggestions:**

I recommend conducting a more comprehensive review of the broader literature, particularly focusing on related work in FL and decentralized computation that addresses DP and noise design techniques. This would help contextualize the proposed approach within existing research and clarify its unique contributions.

In addition, it is important to provide experimental validation to demonstrate the effectiveness of the proposed method. Comparative evaluations against established approaches in FL or decentralized learning would offer valuable insights into its practical benefits

**Other Strengths And Weaknesses:**

No experimental validation is provided in the paper. As a result, it is difficult to assess how the proposed approach enhances privacy in practical settings or how it performs compared to existing methods. Without empirical evidence, the practical effectiveness and advantages of the method remain unclear.

Additionally, the paper offers limited novelty. The core idea of noise cancellation in differential privacy has been previously explored in other contexts, and the transition from full to partial client participation represents only an incremental extension of existing work.

**Questions For Authors:**

No

**Relation To Broader Scientific Literature:**

In terms of the broader scientific literature, the main idea of noise cancellation in differential privacy (DP) is not entirely new. Similar concepts have been applied in other domains, such as in "DPNCT: A Differential Private Noise Cancellation Scheme for Load Monitoring and Billing for Smart Meters."

Beyond the prior work "Private and Federated Stochastic Convex Optimization: Efficient Strategies for Centralized Systems," the primary contribution of this paper lies in extending the analysis from full client participation to partial client participation. However, this incremental advancement offers limited novelty, particularly given that the core technique, noise cancellation, has already been explored in other applications.

**Theoretical Claims:**

It appears correct with explicit assumptions like bounded heterogeneity and Smoothness etc

---

> ### Author Rebuttal · Authors · 2025-04-01
>
> Thank you for the review.
>
> We address all of your concerns and kindly ask you to increase your score accordingly.
>
> ---
>
> **Q: Novelty over the full-participation paper [Reshef and Levy, 2024]**
> **A:** We understand that our approach seems simple in hindsight. However,
> let us highlight that coming up with this idea and analyzing it was *not trivial at all*.
>
> Concretely:
>
> - In **Section 4.1** of our paper, we depict **two natural extensions** of [Reshef and Levy, 2024] that completely fail to achieve optimal guarantees.
> - As shown in our analysis in **Appendices D and E**, the incorporation of **partial participation** and **correlated noise**:
>      - substantially complicates the analysis
>      - introduces challenges far beyond those in [Reshef and Levy, 2024]
>
> - Moreover, our approach is:
>      - the **first to achieve optimal generalization guarantees** in the partial-participation setting
>      - the **first to achieve linear computational complexity** in this setting
>
> - Finally, we believe that:
>
>      - a **simple algorithm is an advantage** rather than a disadvantage
>      - while our approach is simple in hindsight, it was **very challenging to develop and analyze this non-trivial algorithmic idea**
>
> ---
>
> **Q: Novelty over previous work with noise-cancellation, [DPNCT: A Differential Private Noise Cancellation Scheme for Load Monitoring and Billing for Smart Meters]**
> **A:** Please note again that our work is the **first approach to achieve optimal guarantees for DP-FL with partial participation**.
>
> The work you referenced indeed discusses noise cancellation in the context of privacy, and we will gladly cite it in the final version of the paper. However, that paper:
>
>   - does **not consider machine learning training!**
>   - does **not provide any guarantees!** — not even privacy guarantees!
>
> Conversely, our work:
>
>   - discusses **machine learning**, specifically **DP training** in the **federated partial-participation** setting
>   - provides **optimal guarantees** for:
>         - population risk
>         - computational complexity
>         - differential privacy
>   - proposes the use of **noise-cancellation in conjunction with the $\mu^2$-SGD approach**, which is crucial to achieving our guarantees
>   (using noise-cancellation with standard SGD estimates would **fail** to provide optimal guarantees)
>
> ---
>
> **Q: Adding experiments**
> **A:** We have now added experiments that demonstrate the applicability of our approach and corroborate our theoretical guarantees (Logistic regression over MNIST). Please see our response to Reviewer **kvrv**.

---

### Official Review · Reviewer_S51V · 2025-03-14

**Overall Recommendation:** 2

**Summary:**

This paper proposes a differentially private algorithm for federated learning when the untrusted server and with partial participation. The approach is an extension of the previous work of the paper "Private and Federated Stochastic Convex Optimization: Efficient Strategies for Centralized Systems", Roie Reshef, Kfir Y. Levy ICML 2024 which requires full participation.

**Claims And Evidence:**

The claims are supported by the proofs

**Essential References Not Discussed:**

none

**Experimental Designs Or Analyses:**

This is theoretical paper.

**Methods And Evaluation Criteria:**

The approach is only theoretical, and proofs are included, so the methodology is correct

**Other Comments Or Suggestions:**

none

**Other Strengths And Weaknesses:**

The main weakness of this paper is its significant similarity with the previous paper of ICML 2024. The extension to partial participation does not seem very challenging and it is not clear that this should deserve a full ICML paper. The overlap is a bit concerning. For instance, pages 12 to beginning of 16 are nearly identical in both papers, without clear mention of it. It would ease the reader understanding to clear present what is new and what is unherited from previous work.

**Questions For Authors:**

Please comment on the novelty with respect to the ICML 2024.

**Relation To Broader Scientific Literature:**

It seems coherent.

**Theoretical Claims:**

I was not able to check the soundness of the proofs.

---

> ### Author Rebuttal · Authors · 2025-04-01
>
> Thank you for the review.
>
> We address all of your concerns and kindly ask you to increase your score accordingly.
>
> Regarding your comment:
>
> ---
>
> **Q: Novelty over full-participation paper [Reshef and Levy, 2024]**
> **A:** We understand that our approach seems simple in hindsight. However,
> let us highlight that coming up with this idea and analyzing it was *not trivial at all*.
>
> Concretely,
>
> - In **Section 4.1** of our paper, we depict **two natural extensions** of [Reshef and Levy, 2024] that **completely fail to achieve optimal guarantees**!
>
> - Moreover, as you can see in our analysis appearing in **Appendices D and E**, the partial participation and correlated noise:
>
>      - substantially complicate the analysis
>      - are substantially more challenging compared to the analysis of [Reshef and Levy, 2024]
>
> -  In pages 12–16 of the appendix we do have some overlap with [Reshef and Levy, 2024], but these appendices are provided **for completeness**.
> Actually, [Reshef and Levy, 2024] themselves also provide this for completeness, and this part is **not a novelty of their work**.
> We shall highlight this in the final version of the paper.
>
> - Moreover, our approach is:
>
>     - the **first to achieve optimal generalization guarantees** in the partial-participation setting
>     - the **first to achieve linear computational complexity** in this setting
>
> - Finally, we believe that:
>
>     - a **simple algorithm is an advantage rather than a disadvantage**
>     - while our approach is simple in hindsight, it was **very challenging to come up with this non-trivial algorithmic idea and analyze it**

---

### Official Review · Reviewer_kvrv · 2025-03-15

**Overall Recommendation:** 3

**Summary:**

The paper addresses the challenge of ensuring Differential Privacy (DP) in Federated Learning (FL) under partial participation, where only a subset of devices engage in each training round. Existing approaches struggle to extend DP guarantees from full-participation settings to practical FL scenarios with inconsistent availability. The authors propose a novel noise-cancellation mechanism that preserves privacy without degrading convergence rates or computational efficiency. Their method, analyzed in the SCO framework, achieves optimal performance for both homogeneous and heterogeneous data distributions. The results provide a scalable and practical solution for privacy-preserving FL, particularly in large-scale environments.

**Claims And Evidence:**

I think the main theoretical claims are well-supported by the proofs.

**Essential References Not Discussed:**

I think the authors may consider discussing the improvement over the following paper [1], which seems to improve the computation complexity (as well as communication complexity) compared to (Lowy & Razaviyayn,2023)

[1] Gao, C., Lowy, A., Zhou, X., & Wright, S. J. (2024). Private heterogeneous federated learning without a trusted server revisited: Error-optimal and communication-efficient algorithms for convex losses. arXiv preprint arXiv:2407.09690.

**Experimental Designs Or Analyses:**

NA.

**Methods And Evaluation Criteria:**

Due to the theoretical nature of this paper, there are no experiments. The authors did compare their bounds with previous works.

**Other Comments Or Suggestions:**

It seems that the notation $|\mathcal{S}|$ is not defined in the related work section, though it can be infered from the context.

**Other Strengths And Weaknesses:**

Strengths:
+ A nice extension of the previous framework to handle the partial participation scenario, which somehow demonstrates the power of $\mu^2$-SGD approach.
+ The paper is also well-written and easy to follow

Weaknesses:
- Maybe, some experiments could be added, though I understand that this is mainly a theory paper.

**Questions For Authors:**

I have two main comments.

1. It seems that in the lower bound sections, the authors mainly use the results for empirical loss, while later upper bounds are for population loss. I understand one can do a quick reduction from the lower bound for the empirical one to the population one. It would be better to explicitly say it to avoid any confusion.

2. Another comment is about the relationship between the noise cancelation in this paper and the one in the standard binary-tree mechanism (see [1]). This is mainly from my curiosity. Are they related or simply not related at all?

[1] Koloskova, Anastasiia, et al. "Gradient descent with linearly correlated noise: Theory and applications to differential privacy." Advances in Neural Information Processing Systems 36 (2023): 35761-35773.

**Relation To Broader Scientific Literature:**

FL with formal DP guarantees is important, and improving the computational complexity is also meaningful.

**Theoretical Claims:**

I have mainly checked the privacy and convergences of Theorem 5.1 and Theorem 5.2, which appear to be correct to me.

---

> ### Author Rebuttal · Authors · 2025-04-01
>
> Thank you for your positive review and for your comments.
>
> We address all of your comments and kindly ask you to update your score accordingly.
>
> Regarding your comments:
>
> ---
>
> **Q: Comparison to [Gao et al. 2024]**
> **A:** The work of [Gao et al. 2024] indeed provides guarantees for DP-FL learning.
>
> **Regarding guarantees:**
>
> - In the **Full-Participation** setting, [Gao et al. 2024] enjoy *optimal DP generalization* (i.e., population loss guarantees), while requiring a computational complexity which is super-linear in the size of the dataset, i.e., $O(|S|^{9/8})$, where we define $|S|$ to be the overall number of samples used (see Eq. (7) therein).
>   Note that in the notation of [Gao et al. 2024], $|S| = nN$, where they use $N$ to denote the total number of machines, and $n$ to be the size of the data on each machine.
>
> - Comparably, **in this setting we achieve optimal DP generalization while requiring only linear computational complexity in** $|S|$.
>   Note that in the notation of our paper, $|S| = n$, since we denote $n$ as the total number of datapoints used by all machines throughout the training process.
>
> - In the **Partial-Participation** setting, [Gao et al. 2024] obtain *sub-optimal* DP generalization guarantees.
>   Concretely, the total data used in their work is still $|S| = nN$, where $N$ is the total number of machines and $n$ is the number of data points on each machine. This applies even when they are using $M \leq N$ machines in every round (see Alg. 1 and Alg. 2 in their paper).
>
>   The bound they establish in Theorem C.1 (Eq. (17)) shows a convergence rate of:
>   $$O\left(\frac{1}{\sqrt{Mn}} + \frac{\sqrt{d}}{\epsilon \sqrt{M}n}\right).$$
>   Translating this into the size of the dataset $|S| = nN$ implies a bound of:
>   $$O\left(\frac{\sqrt{N/M}}{\sqrt{|S|}} + \frac{\sqrt{d}N/\sqrt{M}}{\epsilon |S|}\right),$$
>   which is suboptimal.
>   Moreover, their approach requires a computation which is superlinear in $|S|$.
>
>   Conversely, in the partial participation case **we obtain optimal guarantees** of:
>   $$O\left(\frac{1}{\sqrt{|S|}} + \frac{\sqrt{d}\sqrt{M}}{|S|}\right),$$
>   and our computational complexity scales **linearly** with $|S|$.
>   Note that in our notation $M$ is the total number of machines.
>
>   We shall add this comparison to the paper.
>
>
> ---
>
> **Q: Stating overall lower bounds, w.r.t. Population loss**
> **A:** Thank you for this suggestion, we will do so.
>
> ---
>
> **Q: Relation of our noise-cancellation mechanism to [Koloskova et al. 2023]**
> **A:** [Koloskova et al. 2023] discusses the idea of adding general correlated noise patterns during DP training. But they:
> - only discuss ERM problems
> - suggest adding correlated noise to FTRL or SGD which use standard gradient estimates
> - do not provide explicit guarantees, and do not discuss partial-participation
>
> Conversely in our work we:
> - discuss and provide population-loss guarantees
> - suggest adding noise-cancellation (and therefore correlated noise) to $\mu^2$-SGD estimates, which is crucial to our guarantees
> - provide explicit bounds for partial-participation
>
> We shall add this discussion to our work.
>
> ---
> **Q: Experiments.**
> **A:** We performed experiments that illustrate the benefit of our approach and corroborate our theoretical findings. See below.
>
> **Experimental Results (MNIST, Logistic Regression)**
> We compare our method (“Our Work”) to Noisy SGD [DL DP] and [Lowy & Razaviyayn, 2023]. All use n=60,000 samples. In Our Work and Noisy SGD, each sample is used only once (single pass). The low accuracy (<70%) is due to privacy noise and single-pass training.
>
> **Privacy-Level Comparison** (fixed m=50, M=100):
>
> | ρ   | Our Work | Time | Noisy SGD | Time | [Lowy & Razaviyayn, 2023] | Time |
> |-----|----------|------|-----------|------|---------------------------|------|
> | 4   | 53.8%    | 13s  | 45.1%     | 9s   | 47.6%                     | 64s  |
> | 8   | 63.7%    | 13s  | 58.9%     | 9s   | 63.3%                     | 282s |
> | 12  | 66.5%    | 13s  | 63.7%     | 9s   | 66.7%                     | 730s |
>
> **Participation-Level Comparison** (fixed ρ=8, M=100):
>
> | m  | Our Work | Time | Noisy SGD | Time | [Lowy & Razaviyayn, 2023] | Time |
> |----|----------|------|-----------|------|---------------------------|------|
> | 20 | 60.8%    | 13s  | 54.9%     | 9s   | 59.7%                     | 114s |
> | 50 | 63.7%    | 13s  | 58.9%     | 9s   | 63.3%                     | 282s |
> | 80 | 63.8%    | 13s  | 57.0%     | 9s   | 65.8%                     | 452s |
>
> **Key Takeaways:**
> - For strong DP requirement (low $\rho$) and partial-participation (low $m$) we consistently achieve better performance with runtimes  comparable to noisy-SGD. The runtime of [Lowy & Razaviyayn] is substantially higher! And in this regime they obtain substantially worse performance.
>
> -For weak DP requirement (high $\rho$) and when  approaching full-participation (high $m$), the accuracy of [Lowy & Razaviyay] improves but the runtime is very high.

---

### Decision · Program_Chairs · 2025-05-01

**Decision:**

Accept (poster)

**Comment:**

The paper proposes a differentially private federated learning method for partial participation scenarios, extending prior work by Reshef and Levy (ICML 2024) on full participation. The novel noise-cancellation mechanism prevents cumulative noise growth, achieving optimal DP guarantees, linear computational complexity, and strong convergence rates. The initial submission was a little weak on experiments and lacked a more thorough discussion about related noise cancellation literature, but these have been addressed during the rebuttal. Still, multiple reviewers perceive the paper as incremental, and I agree that a more careful literature review should be included. The relationship with the ICML 2024 paper by Reshef and Levy must also be explicitly highlighted in the final version to avoid confusion.